# ProactiveLLM: Learning Active Interaction for Streaming Large Language Models

Junlong Tong [1 2]   Yao Zhang [2]   Anhao Zhao [2 3]   Yingqi Fan [2]   Yunpu Ma [4]   Xiaoyu Shen [2]

## Abstract

Standard Large Language Models (LLMs) follow a "read-then-generate" paradigm, causing unnecessary latency and computation. Streaming LLMs alleviate this issue by generating while receiving inputs, but still struggle to decide when to interact with the stream. Existing methods either hard-code interaction timing or rely on costly external alignment signals, such as timing labels, reasoning trajectories, or stronger teachers. In this paper, we propose **ProactiveLLM**, which achieves active interaction by leveraging the model's endogenous states to guide interaction decisions. The model first learns to perceive semantic sufficiency from partial inputs through two complementary training mechanisms: *mask-based streaming modeling* and *synchronized privileged self-distillation (SPSD)*. The former applies monotonic random masking to the input during training, simulating progressively revealed streaming inputs and enabling the model to learn local semantic dependencies from partial-input views. The latter aligns the partial-context student view with a full-context teacher view generated by the same evolving model, allowing privileged full-context evidence to guide the student's understanding under incomplete observations. Together, these mechanisms induce endogenous sufficiency cues *without requiring external teachers or annotations*, providing a versatile foundation for the plug-and-play integration of diverse decision heads. Extensive evaluation across text and speech streaming tasks confirms that ProactiveLLM significantly reduces interaction latency while maintaining quality, validating its capacity for dynamic and active interaction. Code is publicly available at this repository.

[1]Shanghai Jiao Tong University [2]Eastern Institute of Technology, Ningbo [3]Hong Kong Polytechnic University [4]Munich Center for Machine Learning, LMU. Correspondence to: Xiaoyu Shen <xyshen@eitech.edu.cn>.

*Proceedings of the 43rd International Conference on Machine Learning*, Seoul, South Korea. PMLR 306, 2026. Copyright 2026 by the author(s).

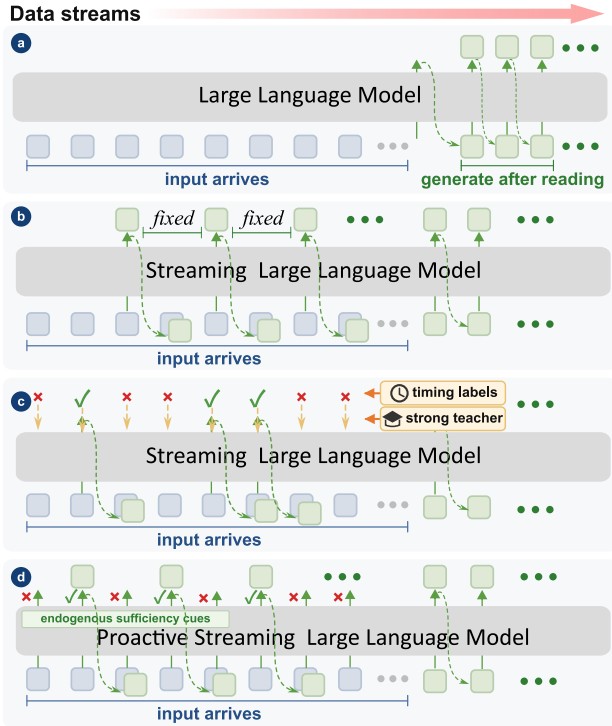

*Figure 1.* (*a*) Standard LLM relys on a "read-then-generate" paradigm. (*b*) and (*c*) Streaming LLM allows input and output to unfold synchronously as streams, but are restricted to fixed decision intervals or or rely on costly external alignment signals.(*d*) *ProactiveLLM* incorporates proactive interaction modeling to adaptively determine the timing of generation with endogenous cues.

## 1. Introduction

Large Language Models (LLMs) (OpenAI, 2023; Team et al., 2023; Yang et al., 2025) inherently exhibit a delayed *batch-processing* characteristic, relying on a "read-then-generate" paradigm that necessitates the ingestion of the full input prior to generation. However, in real-world dynamic streaming scenarios, e.g. audio and video streams, this paradigm suffers from two limitations: (1) *Interaction latency.* The constraint of passively waiting for stream completion severely compromises responsiveness and disrupts real-time interaction. (2) *Cognitive redundancy.* Given that not all generated tokens necessitate global context, forcing every token to attend to the complete input stream incurs

avoidable computational costs and resource inefficiency.

Against this backdrop, the paradigm of streaming LLMs (Tong et al., 2025a; Arora et al., 2025; Cheng et al., 2025; Tong et al., 2026) has emerged, allowing input and output to unfold synchronously as streams. This paradigm promises to significantly reduce interaction latency and mitigate cognitive redundancy. However, this synchronicity introduces a critical challenge: determining the optimal timing to interact with the continuously unfolding input.[1]

Existing methodologies remain largely confined within a *passive adaptation* framework. They either rely on heuristic interaction schedules, such as fixed wait intervals or chunk-wise decoding (Tong et al., 2025a; Chen et al., 2025), or learn generation timing from task-specific alignment supervision (Fu et al., 2025; Chen et al., 2024a), such as input-output alignments, timestamps, or segmentation labels. Although the latter introduces learn- able decision policies, their interaction timing is still an- chored to external annotations and often requires separate data construction or retraining for different tasks, modalities, or latency levels. In both paradigms, generation is optimized to fit an externally prescribed timing pattern, rather than being guided by the model's intrinsic assessment of semantic sufficiency. As a result, models behave as passive followers of predefined rules or supervised alignment signals, limiting their ability to adapt to dynamic streams with fluctuating information density. When partial contexts already contain sufficient semantic cues, such models may still fail to interact before the scheduled or annotated trigger point, leaving the utility of early observations underexploited.

To address these limitations, we propose ProactiveLLM, a streaming framework capable of actively perceiving intrinsic semantic boundaries and making interaction decisions. ProactiveLLM prioritizes learning unified streaming generation capabilities and utilizes learned endogenous states to guide interaction decisions. Specifically, the framework introduces masked streaming modeling and self-distillation, where input token sequences are trained under randomly sampled monotonic visibility masks. This exposes the model to a spectrum of context visibility, ranging from sparse to complete, while full-context logits provide teacher supervision. This training objective therefore compels the model to capture critical monotonic information dependencies and internalize partial-to-full dependency signals, fostering an intrinsic understanding of when partial observations become semantically sufficient. Subsequently, these *endogenous dependency signals and external inputs* serve as inputs for decoupled decision heads, guiding the model's interaction decisions. Moreover, by decoupling capability

learning from decision learning, this architecture enables plug-and-play adaptation to diverse decision heads, allowing the framework to flexibly adapt decoding timing without being limited by specific decoding rules. To validate our approach, we developed both text-streaming and speech-streaming LLMs based on the Qwen series (Yang et al., 2025; Team, 2025) and conducted extensive experiments across tasks including streaming translation, summarization, open-ended QA (short-form QA), and choice QA.

Our ProactiveLLM exhibits an exceptional trade-off between response quality and latency redundancy. Specifically, in non-monotonically aligned tasks such as QA, where decisive evidence may appear at arbitrary positions rather than following the output order, the model retains **97.16%** of the offline performance upper bound while consuming merely **78%** of the input context. Furthermore, it maintains robust performance consistency across diverse backbone models and cross-modal configurations.

In summary, this paper investigates generation timing decisions in streaming LLMs, exploring the boundaries of active interaction capabilities. The main contributions of this paper are threefold: (1) We propose ProactiveLLM, a framework that shifts LLM generation from externally anchored passive adaptation to proactive interaction in streaming scenarios. (2) We introduce mask-based streaming modeling and self-distillation to cultivate partial-to-full dependency and semantic sufficiency signals without requiring any external alignment annotations, providing endogenous cues that enable plug-and-play adaptation to diverse decision heads. (3) We validate our approach across multiple modalities and tasks, including text and speech streams.

## 2. Preliminary

**Definition of Streaming LLMs**   Unlike standard LLMs that adhere to a "read-then-generate" *batch-processing* paradigm, streaming LLMs operate under a "generate-while-reading" *streaming-processing* protocol, as shown in Fig. 1. Formally, given an input stream $\mathbf{X} = \{x_1, x_2, \ldots, x_M\}$ and an output stream $\mathbf{Y} = \{y_1, y_2, \ldots, y_L\}$, the model generates tokens sequentially while the input continues to unfold. The generative process of a streaming LLM can be formalized as decomposing the joint probability $P(\mathbf{Y}|\mathbf{X})$ into a sequence of conditional probabilities:

$$P(\mathbf{Y}|\mathbf{X}) = \prod_{t=1}^{L} P\big(y_t \mid \mathbf{y}_{<t}, \mathbf{X}_{1:\phi(t)}; \theta\big). \qquad (1)$$

Here, $\phi(t)$ denotes the **interaction scheduler**, representing the boundary of the input stream accessible to the model at output step $t$. This function determines the decision timing between input ingestion and output generation. It must satisfy the monotonicity constraint $\phi(t+1) \geq \phi(t)$

---

[1]The model must instantly determine whether the current partial context is semantically sufficient to support valid generation, navigating the trade-off between generation efficiency and quality.

to respect the causal nature of streaming. Notably, when $\phi(t) = |\mathbf{X}|$ for all $t$, the Streaming LLM degenerates into a standard batch-processing LLM.

In existing streaming approaches, $\phi(t)$ is typically defined by static, exogenous rules independent of the input content (e.g., fixed interval methods (Chen et al., 2025; Tong et al., 2025b)). Consequently, the model's generation capability $P(\cdot;\theta)$ is forced to passively fit this rigid schedule.

**Evaluation Metric of Streaming Interaction**   As streaming LLMs necessitate a complex trade-off between redundancy, latency, and quality to process continuous contexts in real-time, we introduce specialized metrics to quantify cognitive redundancy and interaction latency, capturing these dynamic characteristics beyond static generation quality. Specifically, we define **cognitive redundancy** via *read coverage per output token* (RCO):

$$\text{RCO} = \frac{1}{L} \sum_{t=1}^{L} \frac{\phi(t)}{M}, \tag{2}$$

where the $\phi(t)$ denotes the number of input tokens consumed (read) when generating the $t$-th token, $M$ is the input stream length, and $L$ is the output stream length. Intuitively, a lower RCO signifies reduced cognitive redundancy. An RCO of $1.0$ implies that the generation of every token requires the full input stream (i.e., batch processing), whereas a smaller value indicates effective proactivity.

For **interaction latency**, we consider two complementary metrics: relative token-level latency and absolute end-to-end latency. *Relative token-level latency* characterizes how delayed each output decision is with respect to an ideal streaming schedule. We use *average interaction lag* (AIL):

$$\text{AIL} = \frac{1}{L} \sum_{t=1}^{L} \left( \phi(t) - \phi_{\text{ideal}}(t) \right), \tag{3}$$

where $\phi_{\text{ideal}}(t)$ denotes the ideal alignment assuming that the input and output streams unfold uniformly. A lower AIL indicates a more proactive scheduling policy, while a higher AIL reflects a more conservative policy that waits for more context before generation. Complementarily, *absolute end-to-end latency* measures practical wall-clock waiting time from the arrival of the first input unit to the completion of the final output. Let $\tau_{\text{in}}(i)$ denote the arrival time of input unit $x_i$ and $\tau_{\text{out}}(t)$ denote the completion time of output token $y_t$; we define: $T_{\text{abs}} = \tau_{\text{out}}(L) - \tau_{\text{in}}(1)$.

## 3. ProactiveLLM

In this section, we present ProactiveLLM, which reframes the interaction scheduler $\phi(t)$ from a static rule into a model-relevant, content-dependent policy $\phi(t;\theta)$, where the $\theta$ is the model parameters. To implement this, we adopt a decoupled "endogenous training and explicit decision" paradigm,

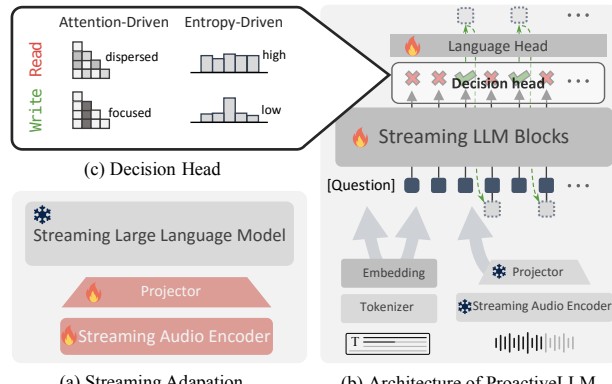

(a) Streaming Adapation   (b) Architecture of ProactiveLLM

*Figure 2.* Streaming LLM backbone and the *ProactiveLLM* architecture. The ProactiveLLM is established on a steaming-adapted LLM with a plug-and-play decision head.

organized as follows: We first introduce the streaming LLM backbone as the foundation. Next, we detail the proactive streaming training framework, which cultivates endogenous boundary perception via masked modeling and anchored self-distillation. Finally, we present the streaming active interaction decision, a plug-and-play mechanism that translates these internal signals into explicit decision actions.

### 3.1. Streaming LLM Backbone

Prior research has demonstrated that group positional encoding effectively adapt batch-processed LLMs to streaming scenarios without requiring architectural modifications (Tong et al., 2025b). Specifically, by decoupling the input and output positional indices, this approach prevents interference during the streaming process while preserving strict input-output temporal alignment. Building upon this foundation, we construct our underlying backbone for both streaming text LLMs and streaming speech LLMs.

Regarding the streaming speech LLM, we utilize a Whisper (Radford et al., 2023) encoder to project audio features into the textual latent space. To ensure end-to-end streaming capability, we impose strict causal masking on the encoder to prevent information leakage. The overall architecture is shown in Fig. 2 (a), with details provided in Appendix B.1.

### 3.2. Proactive Streaming Training Framework

**Masked Streaming Language Modeling**   Drawing inspiration from masked language modeling, e.g., BERT (Devlin et al., 2019), we adapt the masking mechanism to autoregressive streaming scenarios. Unlike BERT, which masks tokens to learn bidirectional representations, our goal is to *simulate the monotonic visibility constraint inherent in streaming.*[2] In a streaming setting, the visible input con-

---

[2]In streaming scenarios, the input content grows progressively over time; consequently, input visibility accumulates in an incre-

text $\mathbf{x}_{\leq\phi(t)}$ available for generating the $t$-th token expands monotonically. During training, we simulate this dynamic availability by applying a randomized causal mask to the future input context. Specifically, for each output token, we randomly mask a portion of the subsequent input tokens. Formally, for each output step $t$, we sample a candidate visible input $\tilde{\phi}(t) \sim Random(M)$ and sort the sequence $\tilde{\phi}$ to the monotonic boundary $\phi$. Here, $M$ is the length of the input stream, and any input token $x_i$ with $i > \phi(t)$ is masked out. In implementation, this process is achieved by masking the output-to-input attention regions, as shown in Fig. 3. Each unique mask matrix corresponds to a specific interaction decision trajectory $\phi$, effectively transforming a static full-input sample into a simulation of a dynamic streaming process. The training objective is to maximize the conditional likelihood of the target tokens given the masked (partial) context, which can be expressed as:

$$\mathcal{L}_{\text{MSLM}} = -\sum_t \log P(y_t \mid \mathbf{y}_{<t}, \mathbf{x}_{1:\phi(t)}; \theta). \quad (4)$$

Plausible interaction decisions should ideally distribute along the monotonic alignment trajectory, reflecting a stable reading pace. While uniform random masking ensures coverage, it inevitably samples degenerate interaction states, e.g., demanding generation with negligible context, that are unreachable during real-world inference. Training on such out-of-distribution noise forces the model to hallucinate rather than ground its predictions. To mitigate this, we constrain the randomness using polynomial-biased multinomial allocation, which which allocates the reading budget across the output tokens during the training process. We define a total reading budget $\mathcal{B} = M$ and allocate it across the $L$ decoding steps by sampling the incremental source consumption $\boldsymbol{\Delta}$ from a multinomial distribution:

$$\boldsymbol{\Delta} = [\Delta_1, \dots, \Delta_L] \sim \text{Multinomial}(\mathcal{B}, \mathbf{w}), \quad (5)$$

where $\mathbf{w} \in \mathbb{R}^L$ is a normalized weight vector (e.g., uniform or polynomially biased) governing the latency preference. The visible mask is derived cumulatively as $\phi(t) = \Delta_0 + \sum_{k=1}^{t} \Delta_k$, where $t \in [1, L]$ and $\Delta_0$ is the prompt length. This constrains decisions near an ideal alignment trajectory, avoiding chaotic patterns from naive random strategies.

**Synchronized Privileged Self-Distillation** While the stochastic masking mechanism enables the model to operate under dynamic streaming constraints, training only with partial contexts introduces a gap between proactive anticipation and full-context verification. The streaming view must predict under incomplete observations, which is essential for low-latency interaction, but this also risks drifting away from the robust full-context distribution learned during

_______________
mental manner and exhibits a monotonic property.

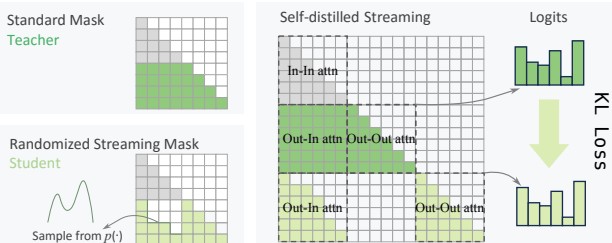

*Figure 3.* (*left*) Standard mask and the masked streaming language modeling in ProactiveLLM, where the streaming boundary is random sampled from a predefined distribution. (*right*) Attention mask for **synchronized privileged self-distillation (SPSD)**, where the "In-In", "Out-In", and "Out-Out" denote attention regions. The teacher and student components denote batch and streaming processing, respectively.

pre-training. To bridge this gap without external teachers or annotated alignment signals, we introduce *synchronized privileged self-distillation* (SPSD). The key idea is to construct a teacher-student relation within the same evolving model under asymmetric context access. Specifically, the streaming mode serves as the partial-context student, while the batch mode serves as the full-context teacher with privileged access to the complete input. Since both views are produced by the current model parameters and optimized jointly during training, the teacher signal is continuously synchronized with the student, rather than being provided by a frozen or external teacher.

During training, the batch mode provides a stable optimization target, representing the ideal completion distribution under full-context conditions. As illustrated in Fig. 3, *we optimize the model for both streaming and batch generation simultaneously*, ensuring that while the model acquires proactive streaming capabilities, its underlying distribution remains anchored to the robust batch baseline. Furthermore, we encourage the streaming mode to align its predictive distribution with the batch logits,[3] using the latter as privileged semantic guidance. Crucially, SPSD does not enforce strict equivalence between the two distributions. The streaming distribution anticipates under partial evidence, whereas the batch distribution verifies under full evidence. Overconstraining the former to match the latter would suppress the proactive behavior we aim to cultivate. Therefore, we incorporate this privileged self-distillation signal as a soft constraint with a minimal coefficient:

$$\mathcal{L}_{\text{distill}} = \lambda \cdot \sum_{t=1}^{L} D_{\text{KL}}\big(P_{\text{batch}}(\cdot \mid \mathbf{x}) \parallel P_{\text{stream}}(\cdot \mid \mathbf{x}_{1:\phi(t)})\big),$$
$$(6)$$

_______________
[3]Given the intrinsic discrepancy between streaming and batch distributions, aligning the full vocabulary would impose incorrect biases from the tail distribution. Hence, we focus the alignment exclusively on the top-$k$ probabilities.

where $\lambda$ is a *small* scaling factor and $L$ is the output token length. Here, $P_{\text{batch}}$ and $P_{\text{stream}}$ represent the distributions from the batch and streaming modes, respectively. This design ensures the model retains the anticipatory variance required for streaming while leveraging the batch logits to prevent catastrophic semantic divergence.

**Unified Training Objective**  To achieve a unified capability of generation and proactive streaming, we formulate the training process as a joint optimization problem. The objective function aggregates the batch loss (batch term), the streaming loss (MSLM term), and the anchored self-distillation loss (KL term). The total loss $\mathcal{L}$ is defined as:

$$\mathcal{L} = \mathcal{L}_{\text{batch}} + \mathcal{L}_{\text{MSLM}} + \lambda \mathcal{D}_{\text{KL}}. \tag{7}$$

Here, each component serves a distinct purpose: $\mathcal{L}_{\text{batch}} = -\sum_t \log P_{\text{batch}}(y_t \mid \mathbf{y}_{<t}, \mathbf{x})$ represents the standard next-token prediction loss on the full input context. This term ensures the model maintains its robust pre-trained knowledge and linguistic coherence, preventing catastrophic forgetting of general capabilities. $\mathcal{L}_{\text{MSLM}}$ (defined in Eq. 4) is the core streaming objective, encouraging the model to interpret and predict under partial-context constraints. $\mathcal{L}_{\text{distill}}$ acts as the stability anchor (as discussed in Sec. 3.2), with $\lambda$ balancing the trade-off between semantic stability and proactive anticipation. By optimizing these objectives jointly, the model effectively learns to read dynamically while staying grounded in its original semantic manifold.

### 3.3. Streaming Active Interaction Decision

Through the random masking modeling and anchored self-distillation described in previous sections, ProactiveLLM has successfully established an endogenous streaming context understanding capability. The model now possesses a latent awareness of whether the current partial context is sufficient for generation. Building upon this internal boundary sensitivity, we introduce an explicit, **plug-and-play** interaction decision head, as shown in the Fig. 2. This unit acts as a gateway controller, leveraging the rich internal representations provided by the frozen LLM to determine the optimal timing for switching between reading the input stream and generating the output stream. Specifically, the decision head monitors the model's intrinsic states, such as token entropy or attention weights, to guide the interaction decision:

(1) Attention-driven head: We monitor the cumulative attention weights allocated to the input stream. A dispersed or negligible attention patterns imply insufficient grounding (prompting a *Read* decision), whereas a sharp focus on specific inputs signals readiness to execute a *Write* decision.

(2) Entropy-driven head: We quantify the model's predictive uncertainty by evaluating the Shannon entropy of the lookahead token distribution. Specifically, let $C_t = (\mathbf{x}_{\leq \phi(t)}, \mathbf{y}_{<t})$

denote the current context, the model probes the next potential token $\hat{y}_t$ and computes:

$$H(P_t) = -\sum_{v \in \mathcal{V}} P(\hat{y}_t \mid C_t) \log P(\hat{y}_t \mid C_t). \tag{8}$$

High $H(P)$ reflects an *information deficit* where the predictive mass is dispersed across multiple hypotheses, necessitating a *Read* to ingest further context. Conversely, low $H(P)$ signifies that the model has reached a high-confidence state with a peaked distribution, authorizing a *Write* decision.

## 4. Experiments

### 4.1. Experimental Settings

**Tasks, Datasets, and Metrics**  We evaluate ProactiveLLM across tasks involving both text and speech input modalities. For text streaming, we select tasks representing monotonically aligned processes, specifically translation evaluated on IWSLT-17 (Cettolo et al., 2017) En→De and En→Fr, as well as non-monotonically aligned processes, including summarization on the Dialogue Summarization dataset, short-form QA on SQuAD (Rajpurkar et al., 2016), and multiple-choice QA on MCTest (Richardson et al., 2013).[4] For speech streaming, the evaluation encompasses Automatic Speech Recognition (ASR) as a monotonically aligned task using the LibriSpeech dataset (Panayotov et al., 2015), and short-form QA as a non-monotonically aligned task using the Spoken-SQuAD dataset (Li et al., 2018). Our evaluation metrics consist of *quality, redundancy, and latency*. While redundancy and latency are defined in Sec. 2, quality metrics are determined according to specific task attributes.[5]

**Models and Baselines**  For text streaming tasks, we employ Qwen2.5-3B-Instruct (Team, 2025) and Qwen3-4B (Yang et al., 2025) as backbone models, applying Supervised Fine-Tuning (SFT) to develop ProactiveLLM. For speech streaming tasks, we utilize Qwen2-Audio-7B-Instruct (Chu et al., 2024) as the backbone, similarly fine-tuned via SFT. To demonstrate the advantages of our proactive interaction strategy, we primarily compare our method against interaction baselines that rely on fixed rules with wait-$k$ intervals (Ma et al., 2019; Tong et al., 2025b).

### 4.2. Text-input Results

Table 1 presents the performance evaluation across four text-input streaming tasks. We establish the "Batch (Full)" setting as the quality upper bound, characterized by full context visibility (RCO=1). However, this comes at the cost of

---

[4]Monotonically aligned tasks exhibit approximate chronological input-output correspondence, while non-monotonically aligned tasks lack this strict temporal alignment. Contrasting them highlights the limitations of fixed-interval methods.

[5]Detailed analysis of metrics is provided in the Appendix D.

*Table 1.* Main experimental results comparing **ProactiveLLM** against static **Fixed Interval Methods** (Wait-$k$) and on the text streaming input tasks. The evaluation spans **Monotonic** (Machine Translation) and **Non-Monotonic** (Summarization and QA) generation tasks. We report task-specific quality metrics (BLEU, ROUGE-L, F1, Accuracy) alongside **Interaction Latency** (AIL ↓) and **Cognitive Redundancy** (RCO ↓). Batch (Full) denote the offline batch-processing upper bound.

| | Monotonic Alignment Streaming Tasks | | | | | | Non-Monotonic Alignment Streaming Tasks | | | | | | | | |
| | MT (en → de) | | | MT (en → fr) | | | Summarization | | | Short-form QA | | | Choice QA | | |
| Methods | BLEU↑ | AIL↓ | RCO↓ | BLEU↑ | AIL↓ | RCO↓ | R-L↑ | AIL↓ | RCO↓ | F1↑ | AIL↓ | RCO↓ | Acc↑ | AIL↓ | RCO↓ |
|---|---|---|---|---|---|---|---|---|---|---|---|---|---|---|---|
| *Qwen2.5-3B-Instruct* | | | | | | | | | | | | | | | |
| Batch (Full) | 27.34 | 8.71 | 1.00 | 36.43 | 8.68 | 1.00 | 36.39 | 70.59 | 1.00 | 74.79 | 77.55 | 1.00 | 88.33 | 204.87 | 1.00 |
| Wait-3 | 14.48 | 2.09 | 0.65 | 12.69 | 2.94 | 0.70 | 12.15 | -7.50 | 0.42 | 8.03 | -27.44 | 0.14 | 43.10 | 3.00 | 0.01 |
| Wait-5 | 15.12 | 3.87 | 0.77 | 14.56 | 4.29 | 0.79 | 12.82 | -5.14 | 0.53 | 8.91 | -25.21 | 0.11 | 45.71 | 5.00 | 0.02 |
| Wait-7 | 16.98 | 5.28 | 0.84 | 23.13 | 5.72 | 0.82 | 13.44 | -2.89 | 0.60 | 11.76 | -22.89 | 0.15 | 47.50 | 7.00 | 0.03 |
| Wait-9 | *21.47* | *6.87* | *0.88* | *25.62* | *7.37* | *0.90* | *14.10* | *-0.63* | *0.64* | *15.14* | *-21.32* | *0.19* | *47.14* | *9.00* | *0.04* |
| **Proactive-Attn** | 20.62 | **5.93** | **0.87** | 30.12 | **6.12** | **0.87** | 32.18 | 49.13 | 0.81 | 71.69 | 59.17 | 0.89 | 83.15 | 151.62 | 0.74 |
| **Proactive-Entr** | **23.62** | 8.36 | 0.88 | **31.83** | 8.50 | 0.88 | **33.89** | 50.77 | 0.81 | **73.48** | 53.14 | 0.75 | **85.83** | 160.46 | 0.78 |
| *Qwen-3-4B* | | | | | | | | | | | | | | | |
| Batch (Full) | 29.41 | 8.71 | 1.00 | 39.01 | 8.68 | 1.00 | 36.45 | 70.70 | 1.00 | 70.75 | 76.53 | 1.00 | 92.38 | 204.98 | 1.00 |
| Wait-3 | 14.91 | 2.55 | 0.67 | 13.59 | 2.94 | 0.69 | 12.92 | -6.61 | 0.45 | 3.53 | -5.13 | 0.44 | 31.79 | 3.00 | 0.01 |
| Wait-5 | 16.58 | 3.81 | 0.76 | 15.03 | 4.29 | 0.78 | 13.38 | -4.85 | 0.56 | 3.74 | -3.46 | 0.45 | 47.50 | 5.00 | 0.02 |
| Wait-7 | 17.07 | 5.36 | 0.83 | 23.49 | 5.74 | 0.84 | 14.15 | -2.23 | 0.63 | 9.83 | -16.81 | 0.30 | 45.36 | 7.00 | 0.03 |
| Wait-9 | *22.97* | *6.92* | *0.89* | *24.33* | *7.29* | *0.89* | *15.04* | *0.12* | *0.67* | *4.66* | *1.03* | *0.48* | *47.98* | *9.00* | *0.04* |
| **Proactive-Attn** | 21.34 | **5.91** | **0.83** | 30.01 | **6.03** | 0.89 | 31.85 | 47.12 | 0.78 | 60.13 | 47.18 | 0.69 | 82.15 | 158.56 | 0.75 |
| **Proactive-Entr** | 18.02 | 7.51 | 0.83 | **32.24** | 7.25 | **0.88** | **34.62** | 48.91 | 0.82 | **62.48** | 49.41 | 0.71 | **83.33** | 154.81 | 0.75 |

*Note: "Fixed Interval" label spans the Wait-3 through Wait-9 rows.*

*Table 2.* Comparison with learning-based interaction baselines on the Qwen3-4B backbone. Qwen3-32B and GPT-5.4 denote the generator used to construct task-specific alignment annotations for the learning-based baseline.

| Latency | Method | MT (En→Fr) | | | Short QA | | |
| | | BLEU↑ | AIL↓ | RCO↓ | F1↑ | AIL↓ | RCO↓ |
|---|---|---|---|---|---|---|---|
| low | Qwen3-32B | 24.12 | 5.76 | 0.83 | 29.84 | **41.55** | **0.56** |
| | GPT-5.4 | **27.18** | 5.93 | 0.84 | 38.12 | 42.68 | 0.58 |
| | **ProactiveLLM** | 26.56 | **5.68** | **0.82** | 48.74 | 41.93 | 0.57 |
| high | Qwen3-32B | 27.62 | 7.41 | 0.89 | 42.88 | **49.63** | **0.69** |
| | GPT-5.4 | **30.74** | 7.58 | 0.90 | 50.21 | 51.02 | 0.72 |
| | **ProactiveLLM** | 30.38 | **7.26** | **0.88** | 58.36 | 49.88 | **0.69** |

*Table 3.* Main experimental results comparing ProactiveLLM on speech streaming input tasks. The evaluation spans Monotonic (ASR) and Non-Monotonic (Spoken short QA) streaming tasks. Note that AIL is measured in seconds (s). For ASR, WER is lower-is-better (↓); for QA, F1 is higher-is-better (↑).

| | ASR Task | | | Spoken Short QA Task | | |
| Methods | WER↓ | AIL (s)↓ | RCO↓ | F1↑ | AIL (s)↓ | RCO↓ |
|---|---|---|---|---|---|---|
| Batch (Full) | 1.60 | 5.10 | 1.00 | 72.15 | 31.02 | 1.00 |
| Wait-3 | 6.85 | 1.45 | 0.58 | 6.45 | 1.45 | 0.28 |
| Wait-5 | 4.20 | 2.25 | 0.69 | 7.12 | 2.25 | 0.25 |
| Wait-7 | 2.15 | 3.05 | 0.78 | 9.50 | 3.05 | 0.29 |
| Wait-9 | 1.95 | 3.85 | 0.85 | 12.85 | 3.85 | 0.32 |
| Proactive-Attn | 1.92 | 3.80 | 0.84 | 69.45 | 23.67 | 0.82 |
| Proactive-Entr | **1.88** | 3.88 | 0.86 | **71.12** | 21.26 | 0.80 |

maximum latency, represented by an AIL of approximately half of input streams length,[6] which stands in contrast to the ideal streaming goal where AIL approaches 0. Our optimization objective is to approach this quality upper bound while keeping redundancy and latency as low as possible.

To further compare with learning-based interaction baselines, we construct text-streaming alignment data for MT and short-form QA using both Qwen3-32B and GPT-5.4 as alignment generators. Due to the high construction cost, we randomly sample 2,000 examples and train all compared models on the same Qwen3-4B backbone. As shown in Table 2, alignment-supervised baselines can slightly improve MT BLEU when generated by GPT-5.4, but they require separate aligned data construction and model training for each latency level. In contrast, ProactiveLLM remains com-

petitive on monotonic MT while yielding lower AIL/RCO. More importantly, on non-monotonic short-form QA, ProactiveLLM outperforms the best learning-based baseline by 10.62 and 8.15 F1 points at the low- and high-latency levels, respectively, showing more robust decision making without external alignment data.

In monotonically aligned tasks, fixed-interval methods like wait-$k$ face a rigid trade-off: improving generation quality necessitates increasing $k$, which inevitably degrades streaming synchronization and increases computational overhead. Breaking this constraint, ProactiveLLM achieves generation quality superior to wait-9 while maintaining lower latency and redundancy than the wait-9 baseline. This suggests that while fixed-interval methods can approach the upper bound, our method offers a superior efficiency-quality balance.

The advantages of ProactiveLLM become significantly more

---

[6] We provide detailed proof and analysis in Appendix D.

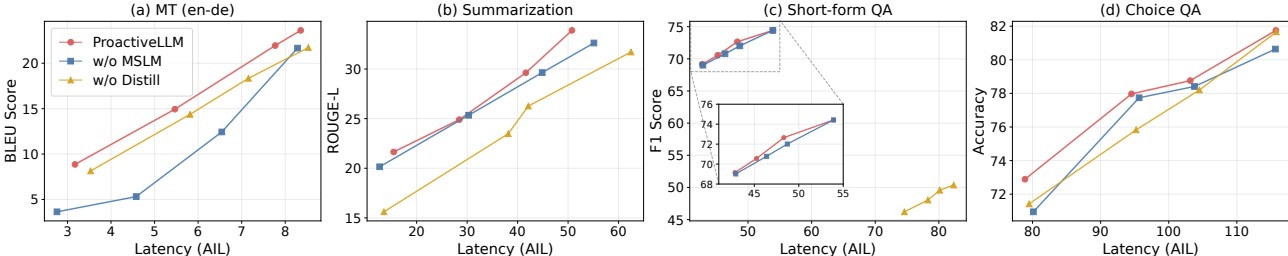

*Figure 4.* **Quality-latency trade-offs across four streaming tasks.** ProactiveLLM (red) consistently defines the optimal Pareto frontier compared to variants without MSLM (blue) or Anchored Distillation (yellow).

pronounced in non-monotonically aligned scenarios where fixed-interval strategies struggle to maintain performance. In such tasks, wait-$k$ methods are frequently compelled to generate outputs based on insufficient context, leading to severe hallucinations. Conversely, ProactiveLLM leverages its active decision-making mechanism to autonomously determine the optimal timing for generation. Taking ChoiceQA as a representative example, ProactiveLLM recovers 97.16% of the upper bound performance while reading only 78% of the input context, resulting in a 21.6% reduction in latency (AIL). This demonstrates that our proactive approach effectively navigates complex context dependencies that rigid, fixed-interval strategies fail to handle.

Similar to wait-$k$, ProactiveLLM also supports flexible trade-offs between latency and quality by adjusting decision thresholds. We explicitly demonstrate this flexibility and the resulting superior Pareto frontier in Ablation Studies.

### 4.3. Speech-input Results

For speech input, taking into account the average human speaking rate, we define the basic decision block as a 400 ms audio segment (corresponding to approximately 20 audio tokens). This segmentation strategy is employed to prevent meaningless decision-making at an overly fine granularity.

Table 3 suggests that the effectiveness of ProactiveLLM extends to speech streaming tasks. In monotonic ASR, our method appears to alleviate the strict latency-accuracy trade-off found in fixed strategies, achieving performance comparable to high-$k$ baselines but with reduced latency. For non-monotonic Spoken Short QA, the advantage is notably more distinct. Given the absence of explicit punctuation in speech streams, fixed-interval methods often face challenges with acoustic boundary detection, potentially leading to premature or delayed responses. In contrast, ProactiveLLM (via Attention and Entropy policies) demonstrates better capability in identifying semantic completion. This allows it to maintain competitive F1 scores while mitigating the synchronization issues often observed in rigid baselines.

### 4.4. Ablation Studies

**Effectiveness of the ProactiveLLM** The core architecture of ProactiveLLM integrates two components: masked streaming language modeling and anchored self-distilled modeling. MSLM is designed to equip the model with intrinsic streaming processing capabilities, enabling it to handle dynamic input streams effectively. Complementing this, anchored self-distilled modeling serves as a stabilizer, ensuring that the model's streaming outputs do not deviate significantly from the full-context backbone.

To rigorously validate each component, we conduct a progressive ablation study with three settings:[7] (1) **ProactiveLLM**: The full model incorporating both MSLM and anchored self-distilled modeling; (2) **w/o Distill**: A variant where anchored distillation is removed, leaving only the streaming modeling; (3) **w/o MSLM**: The baseline backbone model without specialized streaming objectives.

The performance trade-offs across four tasks are illustrated in Fig. 4. The results clearly demonstrate that ProactiveLLM achieves the optimal Pareto frontier across all tasks. As components are systematically removed, the trade-off curves consistently shift towards the bottom-right, indicating a degradation in both latency and quality. Specifically, the removal of self-distillation (*w/o Distill*) leads to a catastrophic performance collapse in reasoning-heavy tasks like Short-form QA, where the F1 score drops significantly while latency increases. Meanwhile, the removal of MSLM (*w/o MSLM*) eliminates the model's ability to maintain high quality under low-latency constraints. These findings empirically validate that both components are essential for achieving robust, high-quality proactive streaming interaction.

**Effectiveness of the KL Coefficient** We further investigate the impact of the KL divergence coefficient ($\lambda$) in the Eq. 6 within anchored self-distilled modeling. As shown in Fig. 5, model performance remains robust across varying coefficient magnitudes.

---

[7]Referring to the mask matrix in Fig. 3, these settings correspond to retaining the complete matrix, the streaming block only, and the batch block only, respectively.

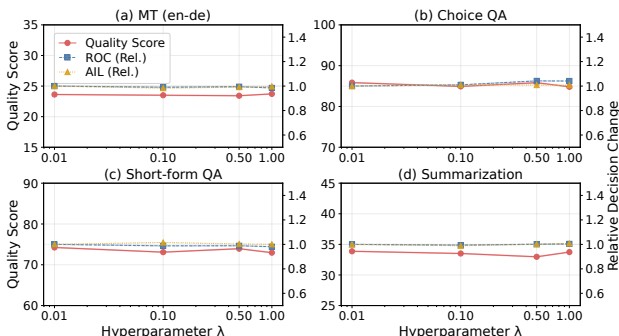

*Figure 5.* Sensitivity analysis of $\lambda$ on Qwen-2.5-3B-Instruct. To visualize stability across different scales, ROC and AIL are normalized relative to their values at $\lambda = 0.01$.

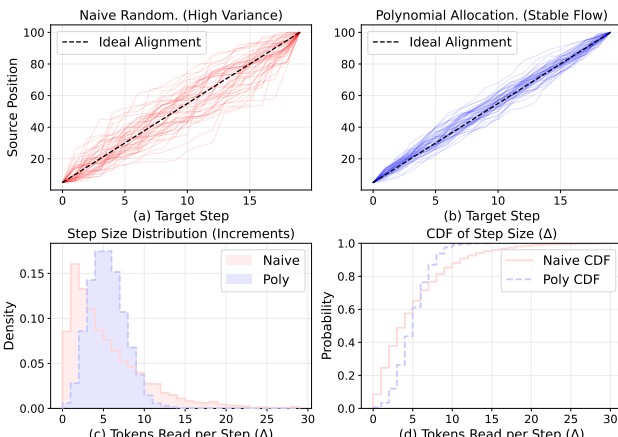

*Figure 6.* Comparative analysis of streaming interaction mask methods during training: Naive Random vs. Polynomial Allocation. The **top row** illustrates the macroscopic alignment trajectories between source and target positions. The **bottom row** quantifies step sizes ($\Delta$) by histograms and CDFs.

*Table 4.* Performance comparison of Wait-$k$ decision: **Trained from Scratch** vs. **Deployed on ProactiveLLM**. Given the effectiveness of Wait-$k$ in monotonic alignment streaming scenarios, we conduct this evaluation on the en-de MT task.

| Methods | Trained from Scratch | | | Deployed on ProactiveLLM | | |
|---|---|---|---|---|---|---|
| | BLEU↑ | AIL↓ | RCO↓ | BLEU↑ | AIL↓ | RCO↓ |
| Wait-3 | 14.48 | 2.09 | 0.65 | **14.65** | 1.95 | 0.62 |
| Wait-5 | 15.12 | 3.87 | 0.77 | **15.42** | 3.65 | 0.74 |
| Wait-7 | 16.98 | 5.28 | 0.84 | **17.05** | 5.25 | 0.83 |
| Wait-9 | **21.47** | 6.87 | 0.88 | 21.30 | 6.85 | 0.88 |

This observation suggests that the explicit KL divergence constraint is not the primary driver of the performance gains attributed to anchored self-distilled modeling. Instead, the fundamental benefit likely stems from the batch term language modeling loss itself, which serves as a high-quality anchor. Crucially, the optimization objectives for the batch term and the streaming term are intrinsically consistent. Since the streaming input can be conceptually viewed as a noise-augmented version of the full batch input, minimizing the standard language modeling loss for both terms simultaneously necessitates an implicit alignment of their representations. Consequently, the optimization process inherently closes the distributional gap between streaming and batch outputs, rendering the magnitude of the explicit KL penalty less critical to the final performance.

## 5. Analysis

**Random Mask Analysis** Masked streaming language modeling is pivotal for ProactiveLLMs to acquire stable endogenous capabilities. To intuitively illustrate the high-noise characteristics of the naive random mask and to validate the effectiveness of our proposed strategy, we compare the statistical properties of polynomial allocation against the naive random baseline, as visualized in Fig. 6.

The top row of the Fig. 6 depicts the alignment trajectories between source and target sequences. The naive random method (red) exhibits extremely high variance with paths frequently deviating from the ideal diagonal, causing the model to either prematurely peek at future inputs or generate excessively without context. In contrast, the polynomial allocation (blue) maintains smoother, stable trajectories, indicating that the budget-based mechanism successfully mitigates such extreme decision-making. This observation is corroborated by the microscopic step size analysis in the bottom row, where the naive method displays a polarization between frequent zero-reads and sudden bursts (long-tail effect). Conversely, polynomial allocation random mask

method yields a low-variance, Gaussian-like distribution centered around the mean budget. This effectively eliminates extreme jumps and establishes a physically consistent continuous data flow, thereby enhancing the robustness of the streaming attention mechanism.

**Generalization Analysis** Our ProactiveLLM framework internalizes intrinsic streaming interaction capabilities during training, thereby enabling seamless plug-and-play support for various downstream interaction strategies, including the attention-driven and entropy-driven policies proposed in this paper. To further substantiate this generalization capability, we extend ProactiveLLM to support rigid rule-based fixed-interval methods. Building on the findings in Sec. 4.2, which indicate that fixed-interval methods are effective for monotonically aligned tasks despite suffering severe degradation in non-monotonic scenarios, we employ the streaming MT task as a testbed to evaluate the adaptation of ProactiveLLM to the Wait-$k$ policy. Tab. 4 demonstrates that directly deploying Wait-$k$ logic on ProactiveLLM achieves performance fully comparable to a specialized Wait-$k$ model

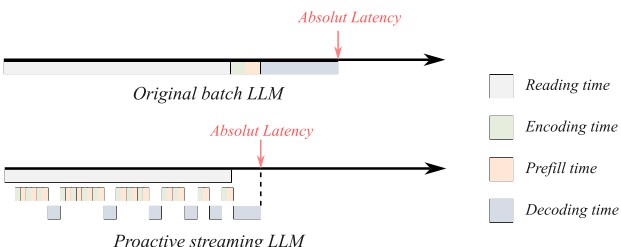

*Figure 7.* Illustration of absolute end-to-end latency. Batch LLMs passively wait until the full stream is received before decoding, making read latency and LLM latency additive. Streaming LLMs overlap input reception, processing, and generation, thereby reducing the final wall-clock latency.

trained from scratch. Notably, ProactiveLLM significantly surpasses the baseline in aggressive low-latency regimes (i.e., when $k$ is small). We attribute this superiority to the model's inherent anticipation capability cultivated from diverse streaming environments. While the baseline is overfitted to a specific, rigid read-write trajectory, our model's broader training scope enables superior resilience under tight latency constraints.

**Redundancy and Latency Analysis**    In this section, we evaluate our model based on token-level redundancy and absolute end-to-end latency. AIL is mainly intended to compare the *relative token-level latency* of different streaming methods by measuring how delayed each write decision is with respect to the ideal diagonal streaming policy $\phi_{\text{ideal}}(t)$. For comparisons with batch LLMs, however, absolute end-to-end latency is more direct: a batch model must first wait until the complete stream is received before decoding, whereas a streaming model can overlap input reception, processing, and generation, as illustrated in Fig. 7. In speech streaming, this read latency is a practical bottleneck because a 10-second utterance inherently takes 10 seconds to receive. On speech tasks, ProactiveLLM (Qwen2-Audio-7B) achieves comparable quality with lower absolute latency than the batch Qwen2-Audio-7B baseline.

We assess the theoretical efficiency of our proactive entropy-based policy by analyzing *Variable Attention FLOPs*. In a decoder-only Transformer (Qwen2.5-3B) with KV Cache, we categorize decoding costs into: (i) Static Overhead (constant FFN and projections) and (ii) Variable Overhead (Self-Attention scaling as $O(L \cdot d \cdot N)$). As FFN cost remains fixed per generated token, we isolate attention FLOPs ($4 \cdot L_t \cdot d \cdot N$) to measure context pruning efficacy.

As shown in Tab. 5, our ProactiveLLM consistently reduces computation across benchmarks, with gains correlating to task-specific information density. Non-monotonic alignment streaming tasks like Multi-choice QA (**34.82%**) and short-form QA (**21.67%**) yield the highest savings by bypassing redundant source context via entropy spikes.

*Table 5.* Cognitive redundancy measured by attention FLOPs analysis on Qwen2.5-3B-Instruct model. $\Delta$ denotes the reduction relative to the full-visibility baseline (batch-processing).

| Task | Batch (GFLOPs) | Streaming (GFLOPs) | $\Delta$ (%) |
|------|----------------|---------------------|--------------|
| MT (en $\rightarrow$ de) | 141.40 | 138.19 | 2.27% |
| Summarization | 1225.01 | 1103.75 | 9.90% |
| Short-form QA | 81.23 | 63.63 | 21.67% |
| Choice QA | 51.00 | 33.24 | **34.82%** |

Conversely, monotonic alignment streaming tasks like MT show minimal reduction (**2.27%**) due to high-fidelity alignment needs. Summarization achieves a moderate reduction (**9.90%**) by filtering conversational noise while preserving core semantic content.

## 6. Conclusion

In this paper, we proposed ProactiveLLM, a framework that advances streaming LLMs from passive adaptation to active interaction. By replacing static heuristic rules with mask-based streaming modeling and self-distillation, our approach empowers models to leverage endogenous states to actively perceive semantic sufficiency. This design effectively decouples generation capability from interaction logic, enabling the plug-and-play integration of diverse decision heads tailored to specific needs. Extensive evaluations across text and speech streaming tasks, ranging from translation to question answering, confirm that ProactiveLLM significantly reduces interaction latency while maintaining generation quality. This work not only addresses the critical challenge of interaction timing but also provides a versatile foundation for building more responsive and efficient real-time AI systems.

## Impact Statement

This research addresses the latency and efficiency challenges of streaming Large Language Models by introducing a framework for active interaction. In real-time streaming scenarios, external teachers, timing labels, and annotated interaction trajectories are often difficult or costly to obtain, limiting the scalability of supervised streaming methods. ProactiveLLM provides a practical alternative by learning endogenous semantic cues from partial inputs without requiring external annotations or stronger teacher models.

By enabling models to decide when to respond based on their own internal states, ProactiveLLM offers a *cold-start solution* for building actively responsive streaming LLMs. This framework can serve as a foundation for future optimization, where reinforcement learning or task-specific preference tuning can further improve interaction timing, response quality, and efficiency. More broadly, our findings contribute to the development of resource-efficient and low-latency LLM systems for real-time AI applications.

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

# A. Related Work

**Streaming Large Language Models**  Research on streaming LLMs primarily focuses on three dimensions: architecture adaptation (Tong et al., 2025b), multimodal extension (Chen et al., 2024a), and interaction policy design (Ma et al., 2019). Architecture adaptation aims to transition models to streaming scenarios, typically utilizing interleaved (Chen et al., 2024a; Du et al., 2024; Liu et al., 2025; Xu et al., 2025b) or grouped (Tong et al., 2025b; Liu et al., 2024a; Tong et al., 2025a) processing mechanisms to handle incremental inputs. In terms of multimodal capabilities, recent studies have expanded streaming processing to diverse tasks, including streaming Chain-of-Thought (CoT) (Tong et al., 2025a; Yakushev et al., 2025; Xie et al., 2025; Chiang et al., 2025), streaming audio (Chiang et al., 2025; Du et al., 2024; Xu et al., 2025a), and streaming video understanding (Chen et al., 2024a; Lee et al., 2025; Lin et al., 2026; Zhang et al., 2026). The design of interaction strategies, determining when to read versus write, remains a core challenge, broadly categorized into rule-based and learning-based approaches. Rule-based methods rely on heuristics such as fixed intervals (defined by words (Ma et al., 2019), sentence (Tong et al., 2025a) or frames (Chen et al., 2025; Lin et al., 2026)) or attention score thresholds (Zhao et al., 2025b; Chen et al., 2024b). Conversely, learning-based strategies seek to optimize policies dynamically, employing techniques such as learnable decision tokens (e.g., in video flows (Chen et al., 2024a)), similarity prediction between streaming and offline states (Zhao et al., 2025b; Chen et al., 2024b), or reinforcement learning-based method (Wang et al., 2025; Xu et al., 2025c) to balance latency and quality. However, existing approaches tend to be specialized for specific tasks and strategies, often tailoring the model to particular decision-making protocols. To address this, we propose a Proactive Interaction mechanism. This approach fosters an intrinsic streaming understanding capability within the model, enabling flexible adaptation to diverse decision heads.

**LLM Distillation**  Knowledge distillation (Hinton et al., 2015) transfers knowledge from a stronger teacher model to a smaller or more efficient student model. In LLM post-training, distillation has been extended from traditional logit matching to response-level imitation, instruction-data distillation (Wang et al., 2023), and rationale or chain-of-thought distillation (Hsieh et al., 2023; Shridhar et al., 2023). Recent on-policy distillation methods (Agarwal et al., 2024; Gu et al., 2024; Zhao et al., 2026b) further reduce the training–inference mismatch by distilling on student-generated trajectories rather than only teacher-forced data. Different from prior LLM distillation methods, ProactiveLLM uses the same model as both student and teacher under asymmetric context access. The teacher observes the full context as privileged information, while the student only sees partial streaming inputs. Instead of relying on a frozen or external teacher, the teacher signal is computed from the current evolving model, keeping the supervision synchronized with the student throughout training. This distinguishes our method from OPSD-style distillation (Zhao et al., 2026c), where student-side trajectories may be sampled on-policy, but the teacher is typically fixed and can become stale relative to the evolving student. Recent on-policy distillation methods (Agarwal et al., 2024; Gu et al., 2024) reduce the training–inference mismatch by distilling on student-generated trajectories rather than only teacher-forced data. OPSD-style distillation (Zhao et al., 2026c) ffurther introduces *output-side* privileged annotations, such as solution rationales or reasoning traces, to provide additional supervision over student-generated trajectories. However, these methods usually depend on external annotations, teacher-generated reasoning traces, or stronger teacher models, which are difficult to obtain in real-time streaming scenarios. In addition, a frozen or separately maintained teacher may become stale relative to the evolving student. Different from these methods, ProactiveLLM *uses the same model as both student and teacher under asymmetric context access.* The teacher observes the full context as privileged information, while the student only sees partial streaming inputs. Instead of relying on a frozen or external teacher, the teacher signal is computed from the current evolving model, keeping the privileged supervision synchronized with the student throughout training.

**Efficient LLMs**  Research on efficient LLMs has primarily concentrated on reducing computational overhead through model-level (Wu et al., 2026b; Egashira et al., 2024; Zhao et al., 2025a; Fan et al., 2025) and token-level (Xiao et al., 2023; Fu et al., 2024; Yang et al., 2024; Fan et al., 2026; Wu et al., 2026a) optimizations. Model-level approaches, such as quantization (Egashira et al., 2024; Liu et al., 2024b) and network pruning (Zhao et al., 2025a; Frantar & Alistarh, 2023; Ashkboos et al., 2024), focus on lowering the intrinsic complexity of the architecture to enhance computational efficiency. Token-level strategies, on the other hand, aim to reduce the data computation volume; techniques like token compression (Fu et al., 2024; Zhao et al., 2026a) and Key-Value (KV) cache optimization (Xiao et al., 2023; Yang et al., 2024; Zhang et al., 2023) are employed to minimize sequence length and memory footprint during inference. Complementing these computation-centric methods, streaming LLMs introduce a novel perspective on efficiency by addressing data stream processing latency (Tong et al., 2025b). Instead of waiting for full static inputs, streaming paradigms enable concurrent processing, reasoning simultaneously with input reception, thereby maximizing responsiveness in real-time scenarios.

# B. Details of ProactiveLLM

## B.1. Backbone of Streaming LLM

**Text LLM**    To evaluate the generalizability and robustness of ProactiveLLM, we utilize two generations of state-of-the-art decoder-only language models as our backbones: Qwen2.5-3B-Instruct and Qwen3-4B. Qwen2.5-3B-Instruct is a highly optimized 3-billion parameter model that serves as our primary baseline for efficiency-focused scenarios. In contrast, Qwen3-4B, with its 4-billion parameter architecture, provides enhanced reasoning capabilities. By verifying our framework across different model scales and generations, we demonstrate that the proactive policy is not dependent on a specific model architecture but is a universal solution for streaming interaction. Following the design of Group Positional Encoding (GPE) (Tong et al., 2025b), we modify the standard transformer architecture to enable seamless streaming. This architecture serves as the prerequisite for our proactive framework by ensuring consistency between batch pre-training and streaming inference.

*(1) Decoupling Positional Indices*: We implement GPE to assign independent positional index streams to the source (input stream) and target (generated output) segments. Specifically, the source tokens maintain their relative positions starting from index 0, while target tokens are assigned a separate positional sequence starting from their own base index. This decoupling prevents positional shifts when new source tokens arrive, preserving the strict input-output temporal alignment required for coherent generation.

*(2) Decoupled Attention and KV Cache*: To address the input-attention mismatch, we apply a specialized causal mask that restricts source tokens from attending to target tokens. Complementing this, we maintain a decoupled KV cache management system. This design allows newly arrived source tokens to be integrated into the existing context incrementally. By avoiding the need to re-encode the entire history at each time step, the model preserves its internal hidden states and ensures the stability of the "wait-or-emit" proactive policy during long-turn interactions. Through these modifications, the text backbone provides the necessary structural support for both pure text streaming and multimodal speech streaming.

**Speech LLM**    For speech streaming tasks, we utilize Qwen2-Audio-7B-Instruct (Chu et al., 2024) as the foundation for ProactiveLLM. The architecture is designed to bridge the gap between continuous audio input and incremental textual generation, ensuring end-to-end streaming capability.

*(1) Integration with Streaming Text Backbone*: The LLM core of our speech model follows the exact architectural modifications described in the Text LLM section. Specifically, it employs GPE and a decoupled KV Cache system (Tong et al., 2025b). This ensures that as audio features are incrementally projected into the textual latent space, the LLM maintains strict temporal alignment and avoids the input-attention mismatch between audio prefixes and generated tokens.

*(2) Causal Transformation of Whisper Encoder*: The original Whisper encoder (Radford et al., 2023) used in Qwen2-Audio-7B-Instruct is designed for offline batch processing, which prevents streaming due to its global look-ahead attention. To adapt it for proactive interaction, we perform a structural transformation: (a) *Causal Convolution*: We modify the initial convolutional layers to use causal padding, ensuring that feature extraction at time $t$ only depends on current and past audio frames; (b) *Causal Self-Attention*: The bidirectional self-attention layers are replaced with strictly causal attention masks to prevent information leakage from the future, allowing the encoder to process audio chunks as they arrive.

While these architectural changes provide the structural basis for streaming, the encoder requires specialized training to recover its representation capability under causal constraints. The specific supervised fine-tuning (SFT) paradigms and training objectives used to align these streaming audio features with the LLM backbone are detailed in Appendix B.2.

## B.2. Model Details

**Two-Stage Training of Speech LLM**    To develop the end-to-end speech streaming capability of PROACTIVELLM, we adopt a two-stage training paradigm to bridge the gap between continuous audio signals and the textual LLM backbone.

*(1) Stage I: Causal Alignment and Audio Projection*: In this stage, we focus on equipping the LLM with the ability to process audio under causal constraints using the LibriSpeech dataset (Panayotov et al., 2015). During training, the LLM backbone is frozen to preserve its pre-trained linguistic knowledge. We only optimize the modified Whisper encoder and the projector layer to ensure that the encoder—now featuring causal convolutions and causal self-attention—can effectively project audio features into the textual latent space without look-ahead information.

*(2) Stage II: Proactive Streaming Adaptation*: In the second stage, we fine-tune the LLM to handle proactive interaction

logic while freezing the audio encoder and projector layer to maintain stable feature extraction. The LLM backbone is trained using the same ProactiveLLM objectives (MSLM and Anchored Self-Distillation) as the text streaming models. This phase ensures the model learns the "wait-or-emit" policy within the speech modality, enabling it to handle the non-linear correspondence of streaming audio.

**Speech Evaluation Tasks and Datasets** Following the two-stage training, we evaluate the speech model on two representative streaming tasks. *Streaming ASR*: We continue to use the LibriSpeech dataset to evaluate the model's ability to perform real-time speech-to-text transcription. *Speech Short-form QA*: We utilize the Spoken SQuAD dataset (**?**) to assess the model's proactive reasoning. In this setup, the audio passage is streamed incrementally, and the model must decide when it has heard enough audio evidence to answer the pre-presented question. This task specifically tests the model's ability to identify critical "trigger" information within a long, non-monotonic audio stream.

### B.3. Implementation and Hyperparameter Settings

We summarize the main model, training, and inference hyperparameters in Table 6. These settings are shared across the reported ProactiveLLM experiments unless otherwise specified.

*Table 6.* Implementation and hyperparameter settings used for ProactiveLLM training and inference.

| Group | Item | Setting |
|---|---|---|
| Model | Distillation top-$k$ | top-100 logits |
| | Polynomial allocation | Multinomial$(B, \mathbf{w})$, where $B$ is the input-stream length and $\mathbf{w} \in \mathbb{R}^L$ is a normalized weight vector over decoding steps. |
| | Allocation weights | Uniform setting, i.e., $w_t = 1/L$ for all decoding steps, where $L$ is the total number of decoding steps and $\mathbf{w} = \{w_1, \ldots, w_t, \ldots, w_L\}$. |
| Training | Batch size | Text: 64; speech: 16 with gradient accumulation over 4 steps. |
| | Learning rate | $5 \times 10^{-5}$ |
| | LR schedule | Linear decay with 3000 linear warmup steps. |
| | Training epochs | 2 |
| | Hardware | $4 \times$ H100 GPUs |
| Inference | Decision threshold | Swept in $[0, 1]$; threshold 1 degenerates to the batch setting; the 0.9 quantile is used for the main reported results. |

## C. Dataset Details

**Text Modality Datasets** We evaluate the text streaming capabilities of ProactiveLLM across four benchmarks. *(1) Simultaneous Machine Translation*: We use IWSLT-17 En-De and En-Fr (Cettolo et al., 2017), derived from TED talks. Each sentence is processed token-by-token to simulate real-world interpretation. *(2) Streaming Dialogue Summarization*: We employ DialogSum (Chen et al., 2021), where the model must distill multi-turn conversations into concise summaries incrementally. *(3) Short-form Streaming QA*: We utilize SQuAD v1.1 (Rajpurkar et al., 2016). The question is provided initially, and the Wikipedia passage is revealed token-by-token. *(4) Multiple-choice QA*: We use MCTest (Richardson et al., 2013) (MC160 and MC500). The model chooses the correct option (A-D) as the story streams.

**Speech Modality Datasets** To assess cross-modal streaming, we evaluate two speech-based tasks. *(1) Automatic Speech Recognition (ASR)*: We use LibriSpeech (Panayotov et al., 2015), comprising 960 hours of read English speech. Audio chunks are fed to the causal encoder to generate transcripts in real-time. *(2) Speech Short-form QA*: We utilize Spoken SQuAD (Li et al., 2018), where the context passages are audio signals while questions remain in text.

**Data Characteristics and Task Monotonicity (Textual Analysis)** As illustrated in Table 7, we provide a brief analysis focused on the textual benchmarks to characterize the variations in information density, which leads to the definition of *non-monotonicity* in proactive interactions:

*(1) Linear vs. Non-Linear Mapping*: In textual tasks such as Sim. MT, the source-to-target length ratio ($R_{S/T}$) remains close

*Table 7.* **Statistics of evaluation datasets.** Source and target lengths are reported in word counts. $R_{S/T}$ indicates the task's information density.

| Task | Dataset | Avg. Source | Avg. Target | Ratio ($R_{S/T}$) |
|---|---|---|---|---|
| Sim. MT | IWSLT17 En-De | 21.4 | 20.2 | 1.06 |
| Sim. MT | IWSLT17 En-Fr | 21.4 | 23.8 | 0.90 |
| Dialogue Sum. | DialogSum | 187.5 | 24.1 | 7.78 |
| Short-form QA | SQuAD v1.1 | 122.3 | 3.2 | 38.22 |
| Multi-choice QA | MCTest | 215.6 | 1.0 | 215.60 |

to 1.0 (e.g., 1.06 for En-De), indicating a near-linear correspondence. Conversely, in Dialogue Summarization and QA, this ratio increases drastically, reaching 215.60 in MCTest. This signifies that the output is a highly compressed distillation rather than a direct mapping.

*(2) The Concept of Non-Monotonicity*: While monotonic text tasks (MT) release information uniformly, QA and Summarization exhibit strong *non-monotonic* characteristics. In these settings, critical "trigger tokens" containing necessary evidence are sparsely distributed. This textual analysis serves as a representative case for understanding why a fixed-latency strategy (like Wait-$k$) often fails in high-ratio scenarios.

*(3) Necessity for Proactive Strategies*: The non-linear correspondence in these text datasets proves that models cannot rely on linear input accumulation. This justifies ProactiveLLM: in non-monotonic streams, an "interaction policy" is required to identify information sufficiency, balancing latency with reliability.

# D. Discussion of Metric

**Quality Metrics**   In addition to the interaction metrics (redundancy and latency) defined in Sec. 2, we employ task-specific quality metrics to evaluate the performance of ProactiveLLM across different modalities and alignment types.

**Monotonic Text Tasks: Translation**   For Simultaneous Machine Translation (Sim. MT) on the IWSLT-17 dataset, we use **BLEU** score to measure quality. This metric calculates the n-gram overlap between the model-generated translation and the human reference. In streaming scenarios, BLEU effectively captures whether the model can maintain local semantic accuracy while processing the source text incrementally.

**Non-monotonic Text Tasks: Question Answering**   We evaluate two types of QA tasks. For **Short-form QA** (SQuAD), we report the **F1-score**, which measures the word-level overlap between the predicted answer span and the ground truth. This is crucial for non-monotonic tasks as it requires the model to correctly identify the start and end of the evidence within a stream. For **Multiple-choice QA** (MCTest), we use **Accuracy** to determine the percentage of questions where the model correctly selects the unique gold-standard option.

**Speech Streaming Tasks: ASR and QA**   In the speech modality, quality is assessed based on transcription and comprehension. **Automatic Speech Recognition (ASR)** on LibriSpeech is evaluated using **Word Error Rate (WER)**, which computes the edit distance (substitutions, deletions, and insertions) between the predicted transcript and the reference. For **Speech Short-form QA** (Spoken-SQuAD), we follow the text-based evaluation and use the **F1-score**. Since the input is audio, the F1-score here reflects both the model's speech-to-text alignment capability and its reasoning logic.

**Non-monotonic Text Tasks: Dialogue Summarization**   Compared to monotonic tasks like ASR or MT, summarization exhibits extreme non-linearity. We use **ROUGE-L** as the primary metric, which focuses on the Longest Common Subsequence (LCS) to evaluate the structural and narrative consistency of the generated summaries.

**Discussion on Metric Reliability**   The results in Table 8 reveal a significant discrepancy between lexical and semantic evaluations in streaming summarization. We observe the following insights:

- **Deceptive Semantic Scores**: Wait-$k$ methods exhibit a "high BERTScore, low ROUGE" trap. While their BERTScore remains near 0.90, their ROUGE-1 and ROUGE-L scores collapse to nearly half of the Batch performance.

*Table 8.* **Extended results for Dialogue Summarization on Qwen-2.5-3B.** We supplement ROUGE-1,ROUGE-L and BERTScore to analyze the correlation between lexical overlap and semantic similarity. The results highlight that ProactiveLLM maintains structural integrity and semantic fidelity, unlike fixed-latency baselines.

| Methods | BERTScore↑ | ROUGE-1↑ | ROUGE-L↑ | RCO↓ | AIL↓ |
|---|---|---|---|---|---|
| **Batch (Full Context)** | 0.9152 | 42.15 | 33.89 | 1.0000 | 70.59 |
| Wait-3 | 0.8985 | 22.45 | 16.21 | 0.1282 | -50.29 |
| Wait-5 | 0.9004 | 24.12 | 18.45 | 0.1454 | -48.36 |
| Wait-7 | 0.9018 | 26.58 | 20.32 | 0.1627 | -46.60 |
| **Proactive-Entropy** | **0.9122** | **41.29** | **33.89** | 0.8139 | 50.77 |

- **Structural Collapse (ROUGE-L)**: The sharp drop in ROUGE-L (e.g., 16.21 for Wait-3) reveals that fixed-latency models produce fragmented and structurally broken sentences. Because they cut context prematurely, they fail to maintain the longest common subsequence required for a coherent narrative.

- **Metrics Mismatch**: This gap suggests that BERTScore can be deceptive for incomplete text. It may reward semantic keyword matches even when the actual sentence structure is destroyed. In non-monotonic tasks like summarization, ROUGE-L is a more reliable indicator of whether the model is actually "summarizing" or just "guessing" words.

- **Stability of ProactiveLLM**: Unlike Wait-$k$, ProactiveLLM is stable across all metrics. It achieves a ROUGE-L of 33.89 (matching the Batch upper bound), proving it preserves both the semantic meaning and the structural flow of the dialogue.

- **Informed Decisions**: The high RCO (0.8139) confirms that ProactiveLLM identifies the precise moment of information sufficiency. This ensures the output is both semantically accurate and structurally sound.

In conclusion, relying only on semantic metrics like BERTScore can hide major quality drops in non-monotonic tasks. The cross-metric consistency, especially the robust ROUGE-L performance, confirms that ProactiveLLM is a much more trustworthy solution for streaming interaction.

## E. Case Study

*Table 9.* Qualitative trace of the proactive interaction process on the MCTest dataset.

---

**Case Study: Proactive Boundary Perception on MCTest**

**System Instruction (Task Prior):**
```
<|im_start>system\n
Answer the following question based on the passage.  Return ONLY a single character:  A, B, C, or D.
Q: What dessert did Joey choose?  Options:  (A) Apple pie (B) Brownies (C) Vanilla shake (D)
Marshmallow cake
<|im_end>\n
```

**Streaming Interaction Trace:**

| Step | Visible Context | Action |
|---|---|---|
| $t = 1$ | Joey went to a baseball game during the winter... | *Read* |
| $t = 45$ | ... After dinner he looked at the menu and wanted dessert... | *Read* |
| $t = 61$ | ... He loved the taste of **marshmallow** so he went with that. | **Write** |
| $t > 61$ | *(Context Skipped): On his way home he thought someone...* | *Terminated* |

**Model Output:**
```
<|im_start>assistant\n D <|im_end>
```

**Proactivity Analysis:** *RCO*: **0.7258** *Skipped*: **27.4%**

---

We perform a qualitative analysis of the interaction process using a representative instance from the MCTest dataset. As illustrated in Tab. 9, the model is first conditioned on the task goal—identifying the chosen dessert—via the system

instruction. During the initial 70% of the narrative, which details baseball and social interactions, the Entropy-driven head maintains a high-entropy state, correctly recognizing an *information deficit*.

The pivotal transition occurs upon encountering the keyword **marshmallow**. At this juncture, the predictive uncertainty collapses, and the model triggers a *Write* action with an **RCO** of 0.7258. Notably, ProactiveLLM terminates the reading process immediately after this evidence appears, effectively bypassing the remaining 27.4% of the context. This confirms the framework's ability to precisely perceive the *minimal sufficient context* required for accurate generation.

Streaming settings reduce interaction latency, but generating from partial information is inherently risky. This limitation arises from the streaming setting itself rather than from a specific model: when the current prefix contains a plausible but incomplete clue, the model may generate prematurely before the decisive evidence arrives. Compared with fixed-interval methods, ProactiveLLM mitigates this issue by using endogenous cues such as entropy and attention to estimate *semantic sufficiency*, rather than relying on a rigid schedule. In real deployments, the risk can be further reduced by adopting a more conservative decision threshold when reliability is more important than minimum latency.

*Table 10.* Failure-mode analysis on a Spoken-SQuAD example. The speech input is shown as text transcription for readability.

---

**Spoken-SQuAD Case: Premature Generation under Insufficient Context**

**Question:** When did Levi's Stadium open?

**Transcribed Streaming Context:**
On May 21, 2013, NFL owners at their spring meetings in Boston voted and awarded the game to Levi's Stadium. The $1.2 billion stadium opened in 2014. It is the first Super Bowl held in the San Francisco Bay Area since Super Bowl XIX in 1985, and the first in California since Super Bowl XXXVII took place in San Diego in 2003.

| Method | Answer | RCO |
|---|---|---|
| Wait-5 | 'May' | 0.01 |
| Wait-9 | 'May 21, 2013' | 0.11 |
| Proactive-Entropy(0.5) | 'May 21, 2013' | 0.11 |
| Proactive-Entropy(0.9) | '2014' | 0.48 |
| Batch(full) | '2014' | 1.00 |

---

Table 10 illustrates this trade-off on a Spoken-SQuAD example. The question asks when Levi's Stadium opened, while the early context first mentions "May 21, 2013" as the date when NFL owners awarded the game to the stadium. Fixed-interval methods and the aggressive Proactive-Entropy(0.5) policy emit this earlier date with low RCO, showing a premature-generation failure under insufficient context. By increasing the decision threshold to Proactive-Entropy(0.9), the model waits until the actual opening year "2014" appears, producing the same answer as the batch model while still using less than half of the full context. This example highlights both the main risk of streaming generation and a practical mitigation path through threshold calibration.

## F. Task Prompting and Interaction Templates

In this section, we provide the detailed construction of prompts used for training and evaluating ProactiveLLM. Our prompting strategy is designed to facilitate *active interaction* by ensuring the model is task-aware before the input stream begins to unfold.

**Task-Aware Instruction Design**   As illustrated in Table 11, we employ a "Goal-First" prompting philosophy. For discriminative tasks such as *Short-form QA* and *Multiple-choice QA*, the specific question and available options are embedded within the system instruction *prior* to the streaming context. This design is critical for our decision policy: by prepending the query, the model's internal states are conditioned on a clear objective, allowing the decision head to detect the exact moment the relevant semantic evidence appears in the input stream.

**Streaming Interaction Format**   To maintain compatibility with modern conversational LLM architectures and facilitate the extraction of endogenous state signals, all tasks are wrapped in a unified ChatML-style format. As detailed in Tab. 11 and

Tab. 12, this structure ensures robust boundary perception by clearly separating task objectives from streaming observations.

**(1) System Block (Task Prior)**: This block establishes the semantic goal by containing task-specific instructions and global constraints (e.g., the directive for MT or Choice QA). Prepending the objective is crucial for our entropy-driven policy, as it allows the model to measure uncertainty relative to a fixed task. **(2) User Block (Incremental Flow)**: This block accommodates the continuously unfolding input stream. To simulate real-world streaming, the context is updated incrementally, forcing the model to operate under a spectrum of context visibility consistent with our mask-based training objective. **(3) Assistant Block (Active Interaction)**: This block represents the transition from reading to generation. For reasoning models like Qwen-3, we implement a closed-loop thinking block to maintain architectural alignment. The timing of emission is governed by the decoupled decision heads.

By strictly decoupling the task intent (System) from the evidence stream (User), ProactiveLLM effectively identifies the *minimal sufficient context* required to resolve the initial *information deficit*.

*Table 11.* Instruction templates for downstream streaming tasks in ProactiveLLM.

| Task | System Instruction Template |
|------|------------------------------|
| **MT** | Translate the following {src_lang} paragraph to {tgt_lang}. |
| **Summary** | Generate a headline for the following article. |
| **Short QA** | Answer the question: {question}
Concisely based on the provided passage. |
| **Choice QA** | Answer the following multiple-choice question using the passage.
Return ONLY a single character: A, B, C, or D. No words, no punctuation.
{question} |

*Table 12.* The streaming interaction format for Short-form QA: A comparison between Qwen-2.5 and Qwen-3.

| *Data Format of Qwen-2.5-3B-Instruct (Standard Short-form QA)* |
|---|
| <\|im_start\|>system\n Answer the question: **What year did construction begin?** Concisely based on the provided passage.<\|im_end\|>\n |
| <\|im_start\|>user\n The Eiffel Tower's construction work began on **January 28, 1887**... *(Streaming context)*<\|im_end\|>\n |
| <\|im_start\|>assistant\n 1887 <\|im_end\|> |
| *Data Format of Qwen-3-4B (with Closed Thinking Block)* |
| <\|im_start\|>system\n Answer the question: **What year did construction begin?** Concisely based on the provided passage.<\|im_end\|>\n |
| <\|im_start\|>user\n The Eiffel Tower's construction work began on **January 28, 1887**... *(Streaming context)*<\|im_end\|>\n |
| <\|im_start\|>assistant\n **\<think\>\n\</think\>**\n 1887 <\|im_end\|> |

