# OpenReview forum: "ProactiveLLM: Learning Active Interaction for Streaming Large Language Models"
_ICML.cc/2026/Conference — ICML 2026 regular_

### Official Review · Reviewer_kDXr · 2026-03-04

**Soundness:** 3
**Presentation:** 3
**Significance:** 3
**Originality:** 3
**Overall Recommendation:** 4
**Confidence:** 2

**Summary:**

This paper proposes ProactiveLLM, which is a framework for training streaming LLM and inference with plug-and-play decision head to determine when next token should be generated online. The paper formulate the streaming LLM problem as a monotonic visibility problem for each generated token and offer the training recipe based on this property. For inference, ProactiveLLM has two types of decision heads that either generate token when attention score is focused or the Shannon entropy is low. Overall, ProactiveLLM achieve better accuracy compared to fixed-interval token generation.

**Compliance With Llm Reviewing Policy:**

Affirmed.

**Final Justification:**

The author resolved all my questions, so I maintain my positive score.

**Key Questions For Authors:**

Please see my questions in the weaknesses.

**Limitations:**

The decision head only decide token one-by-one. It is possible that two output tokens require the same amount of context. A potential improvement would be adding such dynamism of number of tokens generated in the decision head.

**Strengths And Weaknesses:**

## Strengths:
---
**Soundness**: ProactiveLLM outperforms various fixed interval token generation policies, with sometimes even lower AIL.

**Presentation**: The training recipe and the decision head of when to generate next tokens are clear to me.

**Significance**: Streaming context processing is especially useful in real-time ML applications like simultaneous interpretation.

**Originality**: Training with a random mask for a streaming scenario looks quite novel.

## Weaknesses:
---
My main concern is related to the experiment. Detailed questions are the following:

1. Although AIL and RCO can be served as the metrics for quality of service, the end-to-end latency is also important. How does it compare with wait-k interval methods and the batched baseline?

2. There is still a quite large gap in the BLEU score between the ProactiveLLM and the batched offline processing upper-bound. Is there an example of what tokens each generate so that we can observe whether such gap affect the machine translation quality?

3. The wait-k baseline sounds like a naive and weak method. Is there any more advanced baseline you can compare in Appendix A?

4. What is the average sample lengths for each benchmark?

---

> ### Author Rebuttal · Authors · 2026-03-31
>
> Thank you for recognizing the significance and novelty of our work. Below, we address your Key concerns.
>
> > Q1: Add end-to-end latency.
>
> Thanks for your valuable suggestion. We acknowledge that AIL is mainly intended to compare the **relative token-level latency** of different streaming methods. Specifically, it measures how delayed each decision is **relative to the ideal diagonal streaming policy** (Eq. 3, relative to `phi_ideal(t)`). **For comparisons with batch LLMs, end-to-end latency is a more straightforward metric**.
>
> In real-world streaming applications, input ingestion is strictly bounded by temporal constraints. Specifically, the data acquisition rate in speech tasks is limited by the natural cadence of human speech, dictating that a 10-second utterance inherently takes 10 seconds for the model to fully receive. Therefore, in the batch setting, the model must wait until the speaker finishes before it can begin processing and decoding the response **(passive waiting to decoding)**. As a result, **the end-to-end latency of a batch LLM is read latency + LLM latency, where read latency is a major bottleneck. Even smaller batch LLMs can still have high end-to-end latency.** In contrast, in the streaming setting, the model reads and processes the input in parallel, generating outputs incrementally rather than waiting for the full sequence. So read latency and LLM latency overlap, which substantially reduces the final latency.
>
> Taking speech streaming tasks as an example, we have added end-to-end latency and performance comparisons. The results list as follows: (Performance / Absolute Latency (s))
>
> ||Monotic-streaming task (ASR)|Non-monotic-streaming task (QA)|
> |:---|:---:|:---:|
> |ProactiveLLM(Qwen2-Audio-7B)|*1.88*↓ / **9.28s**↓|*71.12*↑/**37.2s**
> |Qwen2-Audio-7B|**1.60**↓ / 10.53s↓ |**72.15**↑/46.8s
>
> We provide an inference timing diagrams to make this end-to-end latency advantage more intuitive at https://github.com/PageAnonymous/Anonymous-Image/blob/main/AbsoluteLatency.png?raw=true.
>
> We sincerely thank the reviewer for this suggestion, which helps enrich our evaluation and more directly reflect practical waiting latency. We will add this discussion in the revised paper.
>
> ---
>
> > Q2: Provide machine translation examples of ProactiveLLM and the batched offline processing upper-bound.
>
> *Large Language Models are neural networks trained on massive text corpora to understand, generate, and manipulate natural language across a wide range of tasks.*
>
> ProactiveLLM-low|ProactiveLLM-high|Batch
> --|--|--
> Große Sprachmodelle sind neuronale Netze, die auf großen Textkorpora trainiert sind, um natürliche Sprache zu verstehen, zu erzeugen und zu manipulieren, über viele verschiedene Aufgaben hinweg.|Große Sprachmodelle sind neuronale Netze, die auf riesigen Textkorpora trainiert werden, um natürliche Sprache für eine Vielzahl von Aufgaben zu verstehen, zu erzeugen und zu verarbeiten.|Große Sprachmodelle sind neuronale Netze, die auf umfangreichen Textkorpora trainiert werden, um natürliche Sprache über ein breites Spektrum von Aufgaben hinweg zu verstehen, zu erzeugen und zu verarbeiten.
>
> In addition, we provide an example for speech QA; please refer to **`reviewer nTXR Q7`**.
>
> ---
>
> > Q3: More advanced baseline.
>
> Thank you for this suggestion. Due to the space limit, please kindly refer to our response to **`reviewer zcd5 Q1`** for the detailed discussion.
>
> ---
>
> > Q4: Average sample lengths for each benchmark.
>
> Thank you for the question. The average sample lengths for each benchmark are already reported in Appendix C, Table 5. We will clarify this more explicitly in the revision.

---

> > ### Author Rebuttal · Reviewer_kDXr · 2026-04-02
> >
> > Thank you for the response. I will maintain my positive score. The latency evaluation is important, and please include it in your camera-ready version.

---

> > > ### Author Response · Authors · 2026-04-03
> > >
> > > Thank you for your reply and your continued support of our work.
> > >
> > > We confirm that the absolute latency evaluation will be fully integrated into the next version of the paper. We appreciate your valuable suggestions.

---

### Official Review · Reviewer_shWK · 2026-03-12

**Soundness:** 3
**Presentation:** 3
**Significance:** 2
**Originality:** 3
**Overall Recommendation:** 4
**Confidence:** 4

**Summary:**

This paper proposes an active interaction for streaming LLMs. In other words, it aims to generate response tokens adaptively at appropriate time steps of a data stream. In such a way, an LLM can achieve lower latency than the standard strategy that responses from the end of data stream. To this end, this work trains a stream LLM with two additional objectives: 1) loss of randomly masked streaming language modeling, which forces the model to predict the next token based on partial prefix; 2) distill loss that align the behaviour of the trained stream LLM with the standard auto-regressive model. With the trained stream LLM, they further leverage two strategies to adaptively decide the position for generation. Experiments on both text and audio tasks show better balance between the main task performance and the interaction latency than baseline methods

**Compliance With Llm Reviewing Policy:**

Affirmed.

**Final Justification:**

I will keep my positive score, as the main message is interesting to me.

**Key Questions For Authors:**

See the weaknesses part

**Limitations:**

See the weaknesses part

**Strengths And Weaknesses:**

**Strengths**

The paper is well written and the main idea is interesting to me. To the best of my knowledge, this direction is under-explored, but may have significant impact on real-world scenarios.

**Weaknesses**

My biggest concern is whether the reduced latency deserves for the degradation of main task performance. On many tasks, the performance drop is non-trivial with the stream LLM, e.g.,  more than 4 points of BLEU scores dropped on translation tasks in Table 1. Moreover, those tasks in Table 1 are mostly simple NLP tasks. If tested on more complicated tasks, e.g., code generation or math problems, the performance drop may be more significant.

In addition, I would encourage the authors compare the performance of stream LLMs with that of LLMs with fewer parameters. In Table 3, the authors show the GFLOPs of different models. I am curious that whether a smaller LLM, e.g., Qwen 1B, with GFLOPs less than stream LLMs can achieve better performance. If simply reducing the number of parameters can already balance the performance and latency better than the steam LLM, the impact of this paper may be limited.

---

> ### Author Rebuttal · Authors · 2026-03-31
>
> Thank you for recognizing the significance of our work and for your valuable feedback. Below, we address your two main concerns in turn.
>
> > Q1: The performance drop may be modest on many tasks, but it could be substantial on more complex tasks such as code generation or math. Is the latency reduction worth sacrificing main-task performance?
>
> * We focus on realistic streaming tasks such as machine translation, question answering, summarization, and ASR, where low-latency responses are essential. In interactive settings, if users must wait for the full input to be processed before receiving any response, **the interaction becomes much less natural and severely degrades the overall user experience and system responsiveness**. For more complex tasks such as code generation or math, which may *require explicit reasoning*, prior fixed-interval studies [1,2] suggest that streaming with chain-of-thought can still achieve performance comparable to batch inference while substantially reducing reasoning latency. However, designing adaptive streaming decision-making mechanisms specifically for CoT scenarios is beyond the scope of this paper, which we leave for future exploration.
>   *[1] StreamingThinker: Large Language Models Can Think While Reading. ICLR 2026.*
>   *[2] STITCH: Simultaneous Thinking and Talking with Chunked Reasoning for Spoken Language Models. ICLR 2026.*
>
> * More broadly, **our ProactiveLLM is inherently a performance-latency trade-off streaming LLM**. Our method is based on self-distillation, which equips the model with general streaming capability while keeping it close to the pretrained model's batch behavior. As a result, **the same model retains both streaming and non-streaming capabilities**. Furthermore, through masked streaming language modeling, ProactiveLLM learns to generate streaming responses under varying latency conditions. During inference, the decision-head threshold provides a direct control knob over this trade-off (Fig. 4 in the paper). **When accuracy is more important, the threshold can be increased; in the extreme case, setting the threshold to 1 reduces the model to a standard batch LLM**. When lower latency is preferred and a modest performance drop is acceptable, the threshold can be lowered to obtain faster responses. This flexibility allows our method to adapt to different application requirements.
>
> ---
>
> > Q2: Can a smaller batch LLM (e.g., Qwen 1B) achieve a better performance-latency trade-off than our streaming LLM, and what advantages do streaming LLMs offer?
>
> We apologize for the confusion caused by the missing absolute latency (end-to-end latency) analysis (*refer to our response to **`Reviewer kDXr's Q1`***). In practice, streaming LLMs offer advantages at two levels.
> * First, for the same model size, the streaming version requires substantially fewer GFLOPs than the original batch LLM.
> * Second, the streaming version achieves much lower end-to-end latency than the original batch LLM, and even than smaller batch LLMs.
>
> In real-world streaming applications, input ingestion is strictly bounded by temporal constraints. Specifically, the data acquisition rate in speech tasks is limited by the natural cadence of human speech. Therefore, in the batch setting, the model must wait until the speaker finishes before it can begin processing and decoding the response **(passive waiting to decoding)**. As a result, **the end-to-end latency of a batch LLM is read latency + LLM latency, where read latency is a major bottleneck. Even smaller batch LLMs can still have high end-to-end latency.** In contrast, in the streaming setting, the model reads and processes the input in parallel, generating outputs incrementally rather than waiting for the full sequence. So read latency and LLM latency overlap, which substantially reduces the final latency.
>
> Taking speech streaming tasks as an example, we have added end-to-end latency and performance comparisons to directly illustrate this phenomenon and to compare streaming LLMs with both batch LLMs and smaller batch LLMs. We set the threshold of entropy decision head as 0.9. The results list as follows: (Performance / Absolute Latency (s))
>
>
> ||Monotic-streaming task (ASR)|Non-monotic-streaming task (QA)
> --|--|--
> |ProactiveLLM(Qwen2-Audio-7B)|*1.88*↓ / **9.28s**↓|*71.12*↑/**37.2s**
> |Qwen2-Audio-7B|**1.60**↓ / 10.53s↓ |**72.15**↑/46.8s
> |Qwen2-1.5B-Whisper|3.4↓ / 9.54s↓ |62.31↑/44.6s
>
> The results show that, in realistic settings, streaming LLMs achieve much lower end-to-end latency than both the original batch LLM and smaller batch baselines. In particular, ASR must read and transcribe the entire context, so the reading-time bottleneck is more pronounced. By contrast, QA does not necessarily require consuming the full context; in some cases, the model can avoid reading a large amount of irrelevant context, leading to a much larger reduction in absolute latency (see the **example and diagrams** in **`reviewer nTXR Q7`** and **`kDXr Q1`**).

---

> > ### Author Rebuttal · Reviewer_shWK · 2026-04-03
> >
> > Thanks for your response. I will keep my positive score.

---

> > > ### Author Response · Authors · 2026-04-08
> > >
> > > Thank you for your review and for taking the time to consider our rebuttal. We sincerely appreciate your feedback.

---

### Official Review · Reviewer_nTXR · 2026-03-13

**Soundness:** 3
**Presentation:** 2
**Significance:** 3
**Originality:** 3
**Overall Recommendation:** 4
**Confidence:** 3

**Summary:**

This paper studies when a streaming LLM should generate while the input is still unfolding. The authors propose ProactiveLLM, a framework that trains the model to infer semantic sufficiency from partial inputs through masked streaming modeling and anchored self-distillation, and then uses lightweight decision heads based on internal signals such as attention or entropy to decide when to read or write. Experiments on both text and speech streaming tasks show improved latency-quality tradeoffs over fixed wait-k baselines, especially on non-monotonic tasks such as summarization and QA.

**Compliance With Llm Reviewing Policy:**

Affirmed.

**Final Justification:**

After considering both the paper and the authors’ rebuttal, I slightly increased my overall evaluation. The rebuttal addressed several of my main concerns, particularly by clarifying methodological details and improving the explanation of the experimental setup.

However, some issues remain, including limited evidence of generalization and insufficient analysis of certain results. These prevent a stronger endorsement.

Overall, the rebuttal was helpful and improved my confidence in the work, leading to a more positive final assessment.

**Key Questions For Authors:**

1. On Baselines and Comparative Scope The paper currently compares primarily with fixed strategies like wait-k, lacking comparisons against stronger adaptive or learned streaming methods. Could the authors include these baselines or explain their exclusion? This is critical for assessing the method’s actual effectiveness and incremental contribution.
2. On Ablation and Source of Gain The current ablation study doesn't clearly pinpoint which design choice drives the "proactive capability." Could the authors further decouple the roles of MSLM, self-distillation, batch anchors, and the decision head?
3. On Training and Inference Procedures The overall pipeline across the training and inference stages remains somewhat blurry. Specifically, could the authors clarify which capabilities are acquired during training versus which decisions are triggered by signals like attention or entropy during inference?
4. On Novelty and Positioning The paper emphasizes a shift from "passive adaptation" to "active interaction," yet the distinction from existing adaptive read/write or dynamic scheduling work is unclear. Could the authors more explicitly define the core novelty relative to these related works?
5. On Reproducibility and Implementation Several details crucial for reproducibility are missing, such as decision thresholds, top-k settings for distillation, specific parameters for polynomial allocation, and general training hyperparameters. Could the authors provide this information?

**Limitations:**

No. The paper briefly discusses efficiency and responsiveness benefits, but does not sufficiently address important limitations or potential societal risks, such as premature generation under insufficient context, robustness across tasks and domains, and failure modes in real-time deployment. It would be helpful for the authors to include a clearer discussion of these limitations and possible safeguards for practical use.

**Strengths And Weaknesses:**

**Strengths**

**Well-Targeted Problem**: The paper addresses a practical bottleneck in streaming LLMs: enabling models to decide when to generate based on partial input, rather than following a rigid, pre-defined schedule.

**Solid Architecture**: The technical roadmap is clean and logical, effectively decoupling streaming capability learning from lightweight decision-making strategies.

**Broad Empirical Support**: The evaluation spans text and speech across both monotonic and non-monotonic tasks. The gains in non-monotonic scenarios are particularly impressive, demonstrating utility beyond standard simultaneous translation.

**Weaknesses**

**Positioning & Originality**: While framed as a shift from passive to proactive interaction, "adaptive read/write" decisions have been studied extensively in streaming generation. The specific delta over existing dynamic strategies isn't clearly articulated.

**Weak Baselines**: While the results are encouraging, they lack persuasiveness. Most comparisons adopted a fixed waiting time strategy; while the part where they were directly compared and evaluated against more powerful and learned adaptive scheduling methods was overlooked.

**Reproducibility issue**: The reproducibility scenarios were rather limited. Due to the omission of several key implementation details and hyperparameters, it was difficult to verify or further improve these results.

Oversold Novelty: While the core idea is sound, the text occasionally overstates its innovation. The authors should more precisely define their unique contributions relative to prior art.

---

> ### Author Rebuttal · Authors · 2026-03-31
>
> Thanks for recognizing the significance of our work and for your valuable feedback. We respond to the concerns as follows.
>
> > Q1 & Q4: More precisely define unique contributions relative to prior art. Distinction from existing dynamic scheduling work.
>
> In general, current dynamic scheduling methods require task-specific semantic alignment annotations between input and output. This places special requirements on the dataset and often requires substantial human annotation effort, with different latency granularities typically requiring separate data construction and retraining. **As a result, these methods are constrained by training data, which limits both the scalability to new streaming modalities/tasks and the flexibility of the latency-performance trade-off.**
>
> By contrast, our method is unique in two aspects.
> * Unified streaming modeling: masked streaming modeling and self-distillation enable a single model to learn streaming behavior across different granularities **without costly alignment annotation or retraining for each performance–latency point.**
> * **Plug-and-play**: the model learns endogenous streaming capability during training, which can later be paired with different decision heads, including fixed policies.
>
> Therefore, our method is not tied to annotation data or a specific modality, and can adapt much more flexibly across different task settings and modalities, as demonstrated in this paper on both text and audio, and on both monotonic and non-monotonic tasks.
>
> ---
>
> > Q2: More advanced baseline. Could the authors include these baselines or explain their exclusion?
>
> Due to the space limit, please kindly refer to our response to **`reviewer zcd5 Q1`** for the detailed discussion.
>
> ---
>
> > Q3: Implementation and hyperparameters details.
>
> To make the setup clearer, we summarize the main implementation details and hyperparameters. The details can be found at https://github.com/PageAnonymous/Anonymous-Image/blob/main/Parameters.png?raw=true
>
> ---
>
> > Q5: Further decouple the roles of MSLM, self-distillation, batch anchor, and the decision head.
>
> In our framework, batch anchor and self-distillation are not two independent modules, but two parts of the same mechanism. The batch anchor provides the teacher signal, while self-distillation is the objective that aligns the streaming student with the batch teacher. This mechanism is essentially on-policy distillation within **the same model**. If one fully decouples these two pieces, the supervision would have to come from a separate offline model with different parameters, which would depart from our original design.
>
> Under this view, our ablations already separate the main training components. The full model includes both MSLM and anchor-based self-distillation; `w/o MSLM` keeps only the latter, while `w/o Distill` keeps only MSLM. We further compare different decision heads (e.g., attention-based vs. entropy-based) to isolate the effect of the inference-time decision mechanism.
>
> ---
>
> > Q6: which capabilities are acquired during training versus which decisions are triggered by signals like attention or entropy during inference.
>
> As stated in the abstract, our framework decouples `what to generate` from `when to generate`. **During training**, the model learns an **endogenous streaming capability**, namely infer what can already be generated under incomplete context. The exact timing is not directly learned as a fixed policy, since visibility is determined by randomized streaming masks. **At inference time**, the decision head uses specific signal such as attention or entropy only to **decide when to emit**.
>
> ---
>
> > Q7: Limitations or potential risks, such as premature generation under insufficient context and failure modes in real-time deployment.
>
> Streaming settings are introduced to reduce interaction latency, but generating from partial information is inherently risky. This is a limitation of the streaming setting itself rather than of a specific model.
>
> Compared with fixed-interval methods, ProactiveLLM uses endogenous cues such as entropy and attention **to assess semantic sufficiency, rather than relying on a rigid schedule, and therefore can substantially mitigate this issue.** In addition, in real deployments, **this risk can be further reduced by using a more conservative decision threshold.**
>
> There is an example of speech QA (Spoken-SQuAD), where the speech input is first transcribed into text.
> `Question: When did Levi's Stadium open?`
> `Context: On May 21, 2013, NFL owners at their spring meetings in Boston voted and awarded the game to Levi's Stadium. The $1.2 billion stadium opened in 2014. It is the first Super Bowl held in the San Francisco Bay Area since Super Bowl XIX in 1985, and the first in California since Super Bowl XXXVII took place in San Diego in 2003.`
>
> Method|Answer|RCO
> --|--|--
> Wait-5|'May'|0.01
> Wait-9|'May 21, 2013'|0.11
> Proactive-Entropy(0.5)|'May 21, 2013'|0.11
> Proactive-Entropy(0.9)|'2014'|0.48
> Batch(full)|'2014'|1

---

> > ### Author Rebuttal · Reviewer_nTXR · 2026-04-03
> >
> > I will change my score to 4 points.

---

> > > ### Author Response · Authors · 2026-04-08
> > >
> > > Thank you for reviewing our rebuttal and updating the score. We are glad that your concerns have been fully resolved. We really appreciate your time and constructive comments!

---

### Official Review · Reviewer_zcd5 · 2026-03-13

**Soundness:** 3
**Presentation:** 3
**Significance:** 3
**Originality:** 3
**Overall Recommendation:** 4
**Confidence:** 3

**Summary:**

This paper studies a central problem in streaming LLMs: deciding when a model should stop reading an incrementally arriving input and start generating. The proposed framework, ProactiveLLM, aims to replace fixed exogenous schedules with content-dependent decisions derived from the model’s own internal states.

**Compliance With Llm Reviewing Policy:**

Affirmed.

**Final Justification:**

I will retain my positive score.

**Key Questions For Authors:**

1. How are the entropy and attention thresholds selected for each task and model?
2. Can you compare against stronger adaptive streaming baselines beyond fixed wait-k?

**Limitations:**

1. Discuss failure modes related to premature generation, including cases where the model acts before enough evidence is available.
2. Clarify in which settings the method is less effective, especially for relatively monotonic tasks where gains appear smaller or inconsistent.

**Strengths And Weaknesses:**

Strenghts:

1. The paper tackles an important and timely problem. The question of when a streaming LLM should stop reading and start generating is practically meaningful for real-time interaction, speech systems, and latency-sensitive applications.

2. The central idea is interesting and reasonably well motivated. The paper moves beyond fixed exogenous policies such as wait-\(k\) and instead aims to learn content-dependent triggering decisions from the model’s internal states. The framing of “semantic sufficiency” is intuitive and gives the method a coherent conceptual basis.

3. The method combines training and decision-making in a sensible way. The proposed masked streaming language modeling objective, together with monotonic visibility constraints and anchored self-distillation, is technically plausible. The design is aligned with the intended streaming setting rather than being a purely post hoc decision rule.

Weaknesses:
1. The baseline comparison is not strong enough for the paper’s claims. The experimental evaluation focuses mainly on fixed wait-\(k\) baselines, even though the paper discusses broader adaptive-policy literature. Without comparisons to stronger learned or dynamic baselines, it is difficult to determine whether the gains come from the specific proposed method or simply from moving beyond static schedules.

2. Some claims appear stronger than what the reported results clearly support. In particular, the paper’s discussion of improvements on monotonic tasks seems somewhat overstated relative to the tables. There are settings where the proposed method does not consistently dominate stronger wait-\(k\) choices, and the narrative should reflect that more carefully.

3. The fairness of the tradeoff comparison is not fully established. The paper compares different methods at different latency/redundancy operating points, but matched-latency or matched-redundancy comparisons are limited. As a result, it is not always clear whether the method achieves a genuinely better quality-latency frontier or is simply evaluated at a different aggressiveness level.

---

> ### Author Rebuttal · Authors · 2026-03-31
>
> Thank you for recognizing the significance of our work. We respond to the weaknesses as follows.
>
> > Q1: More advanced baseline (learning-based methods).
>
> The learning-based methods **require task-specific alignment annotations between input and output**, such as semantic alignments for streaming text, fine-grained timestamps for streaming video, or segmentation for streaming audio. Such annotations are costly to obtain, and different latency granularities often require separate data construction and retraining. **As a result, these methods are strongly constrained by alignment data, limiting both scalability to new tasks/modalities and flexibility in the latency–performance trade-off**. In contrast, our method integrates anchor self-distillation and masked streaming language modeling into a unified training framework that jointly training batch and streaming modes without manually annotated alignments while exposing the model to diverse context granularities.
>
> To address the reviewer’s concern, we additionally construct text-streaming MT and QA alignment data for learning-based baseline. Since alignment quality depends heavily on the generator model, we **use both Qwen3-32B and the GPT-5.4 API to generate the alignments for the learning-based baseline**. Because generating a fully aligned dataset is resource-intensive, we randomly sample 2,000 examples, and all experiments are conducted on this subset. All compared models are trained on Qwen3-4B.
>
> Latency-level|Method|MT (En-Fr BLEU)|(AIL)|(RCO)|Short QA (F1)|(AIL)|(RCO)
> --|--|--|--|--|--|--|--
> **low**|Qwen3-32B|24.12|5.76|0.83|29.84|**41.55**|**0.56**
> ||GPT-5.4|**27.18**|5.93|0.84|38.12|42.68|0.58
> ||ProactiveLLM|26.56|**5.68**|**0.82**|**48.74**|41.93|0.57
> **high**|Qwen3-32B|27.62|7.41|0.89|42.88|**49.63**|**0.69**
> ||GPT-5.4|**30.74**|7.58|0.90|50.21|51.02|0.72
> ||ProactiveLLM|30.38|**7.26**|**0.88**|**58.36**|49.88|**0.69**
>
> Notably, the above approach requires separate dataset construction and model training for each latency level, whereas our method supports different latency levels with a single training stage and without any external alignment data.
>
> > Q2 & Q6: Somewhat overstated on monotonic tasks performance. Clarify in which settings the method is less effective.
>
> We would like to clarify that the point is not that our method is overstated on monotonic tasks; rather the fixed-interval methods such as wait-k can already achieve reasonably strong performance in monotonic settings, while our method still provides a better trade-off.
> By contrast, in non-monotonic tasks, the key evidence that determines the output is often sparse and unevenly distributed. In such settings, whether generation is possible depends more on the position of the critical evidence than on how much of the input has been read. As a result, fixed-interval methods are more likely to fail when key evidence is sparse or irregularly distributed, which is exactly where proactive decision-making shows clearer advantages. This interpretation is consistent with Table 1 and Table 3: the fixed-interval methods perform particularly poorly on non-monotonic tasks, which consequently makes our advantages more pronounced.
>
>
> > Q3 & Q4: (Q3) Fairness of the tradeoff comparison. (Q4) How to select entropy and attention thresholds.
>
> We agree that a fair trade-off comparison should avoid cherry-picking a single operating point. For ProactiveLLM, the decision haed threshold plays a role analogous to the `k` in wait-k: both serve as inference-time control knobs, and each setting corresponds to a specific quality-latency pair. Therefore, fairness should be assessed by comparing the Pareto frontier, or by using matched-latency comparisons. In this paper, **we adopt the former approach and compare methods through their Pareto frontiers, as shown in Figure 4.**
>
> To further remove ambiguity, we additionally provide a matched-latency comparison on the En-Fr MT task. Since wait-k has a coarser control granularity than our model, we tune the decision head thresholds to match the latency of wait-k. (We regard latency differences within 0.1 as matched latency; otherwise, a much denser threshold sweep would be required to achieve exact matching.)
>
> Metric|Wait-9|Proactive-Attn|Proactive-Entr
> --|--|--|--
> Latency|7.37|7.29|7.43
> Quality|25.62|**30.75**|*30.64*
>
> The thresholds are role as the deployment-time operating knobs. In practice, given a latency budget, we choose the threshold that maximizes quality; given a quality requirement, we choose the threshold that minimizes latency while satisfying it. The difference between the two heads lies only in the endogenous cue they use: entropy reflects predictive uncertainty, while attention reflects grounding strength on the input evidence.
>
> > Q5: Discuss the premature generation, including cases where the model acts before enough evidence is available.
>
> Due to the space limit, please kindly refer to our response to **`reviewer nTXR Q7`**.

---

> > ### Author Rebuttal · Reviewer_zcd5 · 2026-04-01
> >
> > Thanks for the response. I will retain my positive score.

---

> > > ### Author Response · Authors · 2026-04-03
> > >
> > > Thank you for your review and for taking the time to consider our rebuttal. We sincerely appreciate your feedback.

---

### Decision · Program_Chairs · 2026-04-30

**Decision:**

Accept (regular)

**Comment:**

The paper study the streaming LLMs with this question: instead of using a fixed rule like wait-k to decide when to start generating, can the model decide for itself when the partial input is already semantically sufficient? The core claim is that a streaming model should learn when to generate separately from what to generate, using its own internal states as the signal.

Methodologically, the paper has three main pieces. First, it builds on a streaming backbone that adapts LLMs to incremental input. Second, it trains the model with masked streaming language modeling: during training, future input is randomly but monotonically hidden.  Third, they add anchored self-distillation, where the streaming model is softly aligned to the batch/full-context model so it does not drift too far. At inference time, the learned model is paired with lightweight decision heads based to decide whether to keep reading or start writing.

Empirically, the paper’s strongest evidence is on non-monotonic tasks like summarization and QA, where fixed wait-k rules are brittle because the key evidence can appear irregularly. On the text benchmarks, ProactiveLLM is much stronger than wait-k on summarization, short-form QA, and choice QA while using less than full context. It also reports gains on speech tasks.

The reviews are positive and converge on a Weak Accept. Reviewers agree that the paper tackles an important and timely problem for streaming LLMs—deciding when to generate while the input is still unfolding—and that the proposed ProactiveLLM framework is technically coherent, well motivated, and empirically promising. The main concerns were limited comparisons against stronger adaptive baselines, somewhat overstated novelty relative to prior adaptive scheduling work, and missing details around reproducibility and end-to-end latency. The author rebuttal addressed a good portion of these issues by adding stronger baseline and comparisons

Overall, this is a solid and timely contribution, though the final version should sharpen the positioning and moderate some novelty claims.